# Learning to Label: A Reinforced Self-Evolving Framework for Semi-supervised Referring Expression Segmentation

**Runlong Cao** [1]   **Ying Zang** [* 2]   **Chuanwei Zhou** [3 4]   **Tianrun Chen** [5]   **Tong Zhang** [1]   **Zhen Cui** [6]   **Chunyan Xu** [* 1]

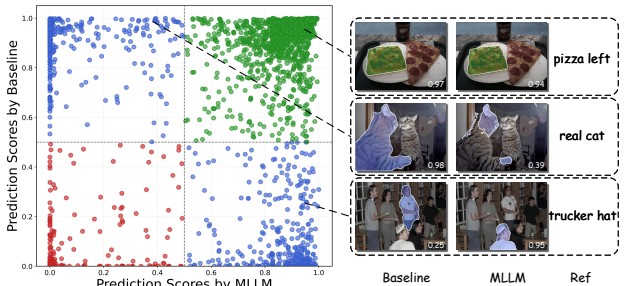

*Figure 1.* Joint distribution of prediction scores: Baseline (w/o MLLM) *vs* MLLM on the RefCOCO dataset.

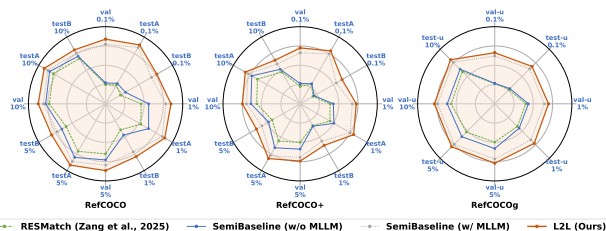

*Figure 2.* Performance comparisons on RefCOCO, RefCOCO+, and RefCOCOg datasets at different label rates.

## Abstract

Semi-supervised referring expression segmentation (SS-RES) aims to achieve precise pixel-level language grounding under limited annotation, yet suffers from limited supervision and unreliable pseudo-labels when exploiting unlabeled image–text pairs. In this work, we propose *Learning to Label*, a reinforced self-evolving framework (L2L) that casts pseudo-label construction as a learnable decision-making process. To build foundational understanding, we leverage a multimodal large language model to extract semantic–spatial priors, which are instantiated as initial soft segmentation proposals and elevated—together with textual cues—into learnable guidance signals that condition a hierarchical segmentation network. To ensure stable learning, a reinforced pseudo-label selection is further formulated as an exploratory decision process that adaptively rewards high-utility pixel-level supervision based on multimodal priors and model predictions. This reinforced self-evolving loop enables joint optimization of the segmentation model and pseudo-labels, progressively enhancing label reliability under sparse supervision. Extensive experiments on RefCOCO, RefCOCO+, and RefCOCOg datasets demonstrate improvements over existing methods, validating its effectiveness and generalization. Code is available at: https://rainloongcao.github.io/Learning2Label/.

## 1. Introduction

Referring Expression Segmentation (RES) aims to segment, at the pixel level, the target object specified by a natural language expression, thereby establishing fine-grained alignment between vision and language (Hu et al., 2016; Mao et al., 2016). As a core task that tightly couples visual grounding and language understanding, it has been widely used in interactive image editing (Zhao et al., 2024), instruction-driven human–robot collaboration (Shridhar et al., 2023), and autonomous driving (Lin et al., 2024). Despite recent progress in multimodal fusion and cross-modal reasoning architectures (Ding et al., 2021; Yang et al., 2022), its performance still relies heavily on large-scale fully supervised training: each image and expression pair requires dense pixel-level masks and expression–object alignment annotations. However, pixel-wise mask annotation is expensive, and expression alignment further amplifies labeling cost, making it particularly difficult to build large-scale RES datasets in long-tail scenarios or domain-specific applications. Therefore, semi-supervised RES (SS-RES) has emerged as a natural direction: training with a small labeled set together with abundant unlabeled image–text pairs to reduce annotation dependence and improve scalability.

---

\* Corresponding authors. [1]School of Computer Science and Engineering, Nanjing University of Science and Technology [2]School of Information Engineering, Huzhou Normal University [3]School of Artificial Intelligence, Nanjing University of Posts and Telecommunications [4]National Key Laboratory of Tibetan Language Intelligence [5]Zhejiang University [6]Beijing Normal University. Correspondence to: Ying Zang <02750@zjhu.edu.cn>, Chunyan Xu <cyx@njust.edu.cn>.

*Proceedings of the 43rd International Conference on Machine Learning*, Seoul, South Korea. PMLR 306, 2026. Copyright 2026 by the author(s).

Most existing semi-supervised segmentation methods are based on pseudo-labeling and consistency regularization (Sohn et al., 2020; Yang et al., 2023), where high-confidence predictions on unlabeled samples are treated as supervision. While effective for semantic segmentation, directly transferring this paradigm to RES is non-trivial. RES is highly sensitive to referential ambiguity, occlusion, and long-range contextual reasoning, which often leads to boundary uncertainty and unstable pseudo-masks. Consequently, prior work (Zang et al., 2025) typically relies on a fixed confidence threshold to filter pseudo-labels, yet such heuristics are brittle: an overly strict threshold discards many informative unlabeled samples, while an overly loose threshold introduces erroneous masks and quickly contaminates training. Recently, the rise of multimodal large language models (MLLM) (Liu et al., 2023b; Bai et al., 2025) and general-purpose segmentation models (Kirillov et al., 2023; Chen et al., 2023; Ravi et al., 2025; Carion et al., 2025) provide new sources of supervision for SS-RES. MLLM offers stronger semantic understanding and referring reasoning, and SAM-style segmentors can produce structured region candidates under weak or even no supervision; such external priors make it possible to generate higher-quality pseudo-labels (Yang et al., 2024). However, these external priors are not always reliable: under occlusion or ambiguous expressions, MLLM may hallucinate or suffer grounding drift, and SAM-style proposals may over-segment or be attracted to distractors. This motivates a practical question: to what extent do MLLM agree with the segmentor's own predictions on unlabeled image–expression pairs?

As shown in Figure 1, we visualize the joint distribution of prediction scores (x-axis: MLLM; y-axis: Baseline), and compute the sample-level confidence for each unlabeled image–expression pair based on both the baseline model and MLLM. Specifically, the scores are defined as the spatial average of a certainty map derived from the foreground probability. A pronounced confidence mismatch is observed between the two sources: a substantial fraction of samples lie in off-diagonal regions, corresponding to cases where the prior is confident while the model is not, or vice versa. For example, in the case of "trucker hat", MLLM's confidence is 0.95, while the baseline's confidence is only 0.25; conversely, for "real cat", the baseline's confidence is as high as 0.98, while MLLM's confidence is only 0.39. Such systematic and sample-dependent disagreement makes the fixed-threshold pseudo-labeling approach brittle: overly conservative thresholds discard informative, learnable samples, while overly permissive thresholds introduce noisy supervision that can quickly contaminate the training process. Based on these observations, we propose an uncertainty-aware mechanism that fuses external priors with model predictions at the pixel level, along with a dynamic, sample-adaptive pseudo-label selection strategy to balance

precision and coverage throughout the training process.

To address the above challenges, we propose *Learning to Label* (L2L), a reinforced self-evolving framework for SS-RES that formulates pseudo-label construction on unlabeled data as a learnable decision-making process. L2L establishes a closed-loop paradigm: multimodal priors, pseudo-label selection, and the segmentor co-evolve, yielding progressively improved pseudo-label reliability on unlabeled data. Specifically, we introduce Semantic-spatial Prior with MLLM, which leverages a parameter-frozen MLLM to infer semantic grounding cues for unlabeled image–text pairs and uses them to prompt SAM2 to produce dense soft segmentation proposals with uncertainty preserved. Building on these priors, we design the Self-Evolving Segmentation Module (SESM), which elevates multimodal priors and textual cues into learnable guidance signals to condition a hierarchical segmentation model, progressively transforming coarse priors into refined, context-aware predictions. Finally, to stabilize self-training under sparse supervision, we propose a Reinforced Pseudo-Label Exploration (RPLE), which formulates pseudo-label acceptance as a reinforced, sample-dependent decision process and adaptively rewards high-utility pixel-level supervision based on multimodal priors and model predictions, thereby dynamically balancing precision and coverage throughout training. As reported in Figure 2, extensive experiments on RefCOCO, RefCOCO+, and RefCOCOg datasets demonstrate that L2L consistently outperforms existing semi-supervised and baselines under various labeling ratios. Notably, with only 0.1% labeled data, our method remains competitive with zero-shot counterparts, validating the effectiveness and robustness of the proposed framework. In summary, the primary contributions of this work are three-fold:

- We propose *Learning to Label*, a reinforced self-evolving framework (L2L) for SS-RES that formulates the pseudo-label construction to a learnable decision-making process, enabling more reliable mining of pixel-level supervision under sparse annotations.

- We introduce MLLM-driven semantic–spatial priors to initialize and guide soft segmentation, and further design a reinforced pseudo-label selection that adaptively identifies high-utility pixel-level supervision based on multimodal priors and model predictions.

- We conduct extensive experiments on three public benchmarks, demonstrating consistent improvements over existing methods across different supervision regimes and validating the effectiveness and generalization of the proposed approach.

## 2. Related Work

**Referring Expression Segmentation:** Referring Expression Segmentation (RES) is a multimodal task that requires

localizing and segmenting objects in images based on free-form natural language descriptions (Hu et al., 2016). This task poses unique challenges as it demands not only visual understanding but also reasoning about linguistic attributes, spatial relationships, and contextual cues (Yu et al., 2018). The field has progressed from early fusion-based methods that simply concatenate visual and textual features (Liu et al., 2017; Li et al., 2018) to Transformer-based architectures that enable dense cross-modal interactions (Ding et al., 2021; Yang et al., 2022). Notably, recent works have explored diverse fusion strategies, including query-based decoding (Kim et al., 2022), pixel-level alignment (Wang et al., 2022), and hierarchical reasoning (Huang et al., 2020). The scope of RES has also been broadened to handle more realistic scenarios, such as expressions referring to multiple objects or no objects at all (Liu et al., 2023a). However, a common limitation shared by existing methods is their reliance on exhaustive pixel-level annotations paired with detailed language descriptions, which restricts their scalability to new domains and practical applications.

**Semi-Supervised Segmentation:** Semi-Supervised Learning (SSL) aims to jointly exploit unlabeled data and a small set of labeled samples to alleviate data scarcity caused by expensive annotations. FixMatch (Sohn et al., 2020) generates pseudo-labels from weak augmentations and supervises strongly augmented views, typically filtering pseudo-labels with a fixed confidence threshold. Pseudo-label quality is critical for SSL, yet filtering often relies on a fixed confidence threshold. Such a static strategy may discard informative hard samples and induce training bias (Zhang et al., 2021). To address this issue, prior work proposes dynamic thresholding mechanisms (Zhang et al., 2021; Xu et al., 2021; Liu & Liu, 2025). Different from these, we further view threshold selection as a reinforcement learning decision process and adopt richer state representations. Recently, SSL has also been extended to RES, giving rise to SS-RES (Yang et al., 2024; Zang et al., 2025). RESMatch (Zang et al., 2025) extends consistency regularization to RES, while SemiRES (Yang et al., 2024) leverages SAM to refine pseudo-labels. Despite these advances, pseudo-label filtering in this line of work is still driven by fixed confidence thresholds, and the selection stage does not explicitly model sample difficulty for adaptive threshold decisions.

**Multimodal Large Language Model and Foundation Priors:** Large foundation models are increasingly reshaping the overall pipeline of referring segmentation and open-world segmentation (Zhu et al., 2025a; Zang et al., 2026). Segment Anything Model (SAM; (Kirillov et al., 2023)) and its successor SAM2 (Ravi et al., 2025) provide strong class-agnostic mask priors that can be used for proposal generation or mask refinement. SemiRES (Yang et al., 2024) serves as a representative SS-RES method by incorporating SAM as an external mask prior to refine pseudo-labels.

Open-world grounding models (e.g., GroundingDINO; (Liu et al., 2024)), as well as grounding-to-mask systems that combine grounding with SAM (e.g., Grounded-SAM), further enable a more powerful localization-to-segmentation workflow in broader scenarios (Shen et al., 2024). More recently MLLM have advanced reasoning-based segmentation: LISA connects language tokens with SAM to produce masks (Lai et al., 2024), and a line of subsequent work explores MLLM-enhanced universal segmentation or referring-aware segmentation, such as GSVA (Xia et al., 2024), InstructSeg (Wei et al., 2025), CPCF (Zhu et al., 2025c), SegLLM (Wang et al., 2025b), Popen (Zhu et al., 2025b). In this work, we condition the segmentation network on multimodal semantic–spatial priors and pose pseudo-label filtering as a reinforcement learning decision process, enabling sample-adaptive supervision.

## 3. The Proposed Method

### 3.1. Overview

How to effectively leverage imperfect MLLM priors together with the model's own predictions to generate high-quality pseudo labels remains a critical challenge. Fixed-threshold pseudo-labeling is brittle in the presence of systematic discrepancies between the two sources. To address this issue, we propose *Learning to Label* (L2L), which upgrades pseudo-label construction from static threshold truncation to a learnable decision-making process, allowing the segmentor and pseudo-label quality to co-evolve in a closed loop. As shown in Figure 3, L2L first leverages a parameter-frozen MLLM to generate coarse grounding cues $b_i$, which are then used to prompt SAM2 to obtain a soft segmentation prior $P_i^{\dagger}$. Next, SPM performs uncertainty-aware calibration between $P_i^{\dagger}$ and the weak-view prediction $P_i^w$, yielding a more reliable fused prior $\tilde{P}_i^{\dagger}$ for pseudo supervision and structural guidance. Building upon the calibrated prior, SESM injects semantic and structural guidance via a hierarchical, stage-adaptive gating mechanism to progressively refine representations and predictions. Finally, RPLE adaptively selects trustworthy pixel regions for pseudo supervision conditioned on the sample state, thereby dynamically balancing pseudo-label precision and coverage throughout training and mitigating noise accumulation.

We follow the weak-to-strong consistency paradigm. Our goal is to learn a segmentation network $S_\theta$ by jointly leveraging $\mathcal{D}_l$ and $\mathcal{D}_u$, such that it produces pixel-wise foreground probability predictions for any image–expression pair. Given an unlabeled set $\mathcal{D}_u = \{(I_i^u, T_i^u)\}_{i=1}^{N_u}$ and a labeled set $\mathcal{D}_l = \{(I_i^l, T_i^l, Y_i^l)\}_{i=1}^{N_l}$ with $N_u \gg N_l$, we construct weakly and strongly augmented views for each unlabeled sample and obtain the corresponding predictions: $P_i^w = S_\theta(\mathcal{A}_w(I_i^u, T_i^u)), P_i^s = S_\theta(\mathcal{A}_s(\mathcal{A}_w(I_i^u, T_i^u)))$. Here, $\tilde{P}_i^{\dagger}$ provides a more reliable pool of pseudo-supervision candidates, while RPLE further identifies learnable regions from

it to impose consistency-based supervision on the strong-view prediction, enabling stable semi-supervised optimization (Details of the SS-RES setting are in the Appendix A.1).

## 3.2. Self-Evolving Segmentation Module (SESM)

RES demands both semantic grounding to the expression and accurate spatial disambiguation, so as to produce boundary-precise target masks in complex scenes. Moreover, although MLLM can provide strong external priors, their grounding cues may be noisy or even hallucinated; therefore, such priors should be exploited in a learnable and calibrated manner. In SESM, we inject the pixel-wise prior $P_i^\dagger$ as conditioning into a hierarchical segmentor via stage-adaptive gated modulation. As training progresses, $P_i^\dagger$ is continuously updated based on the increasingly improved weak prediction $P_i^w$, providing multi-level guidance for progressive refinement. Let $\mathcal{V}_j \in \mathbb{R}^{N_j \times C_j}$ denote the visual representation at stage $j$ of a hierarchical encoder. We consider two complementary guidance sources: a semantic field $\mathcal{S}$ produced by the text encoder, and a stage-aligned structural field $\mathcal{G}_j$ derived from the pixel-wise prior $P_i^\dagger$ (where $i$ indexes training samples). Specifically, we obtain $\mathcal{G}_j$ via a stage-alignment operator applied to $P_i^\dagger$, such that $\mathcal{G}_j$ is strictly resolution-aligned with the stage-$j$ visual features. Therefore, $\mathcal{G}_j \in \mathbb{R}^{N_j \times C_j}$ is aligned with $\mathcal{V}_j$ in spatial resolution (with $N_j = H_j W_j$). Instead of static fusion, we synthesize a unified conditioning signal $\mathcal{M}_j$ via dynamic modality gating:

$$\mathcal{M}_j = \alpha_j \cdot \Psi_{sem}(\mathcal{V}_j, \mathcal{S}) + \beta_j \cdot \Psi_{geo}^{(j)}(\mathcal{V}_j, \mathcal{G}_j), \quad (1)$$

where $\Psi_{sem}$ is implemented with standard cross-modal attention to retrieve linguistic cues conditioned on $\mathcal{V}_j$. The scalars $\alpha_j$ and $\beta_j$ are learnable stage-wise gating factors that adaptively balance high-level semantic disambiguation and low-level structural refinement across the hierarchy.

To match the structural interaction with the intrinsic abstraction level of visual features, we define $\Psi_{geo}^{(j)}$ as a stage-dependent piecewise function. Based on our 4-stage encoder, we decouple the interaction into strictly aligned local projection for shallow stages and global relational reasoning for deep stages:

$$\Psi_{geo}^{(j)}(\mathcal{V}_j, \mathcal{G}_j) = \begin{cases} \mathcal{W}_p(\mathcal{G}_j), & j \in \{1, 2\}, \\ \text{Softmax}\left(\frac{(\mathcal{V}_j \mathbf{W}_Q)(\tilde{\mathcal{G}}_j \mathbf{W}_K)^\top}{\sqrt{d}}\right)(\tilde{\mathcal{G}}_j \mathbf{W}_V), & j \in \{3, 4\}, \end{cases} \quad (2)$$

where $\mathcal{W}_p$ denotes a learnable linear projection that preserves strict spatial correspondence in shallow stages. $\mathbf{W}_Q, \mathbf{W}_K, \mathbf{W}_V$ are projection matrices for cross-attention to enable global relational reasoning in deep stages, and $\tilde{\mathcal{G}}_j$ is the flattened sequence representation of the structural field. $d$ is the feature dimension used in the attention computation.

To robustly incorporate $\mathcal{M}_j$ while preserving the pre-trained

feature distribution, we apply a gated residual update:

$$\mathcal{V}_j' = \mathcal{V}_j + \sigma(\mathcal{W}_g(\mathcal{V}_j)) \odot \mathcal{M}_j, \quad (3)$$

where $\mathcal{V}_j'$ is the updated feature, $\sigma(\cdot)$ denotes the sigmoid function, $\odot$ is element-wise multiplication, and $\mathcal{W}_g$ predicts token-wise conditioning strength to dynamically regulate the influence of the guidance signal. Meanwhile, $P_i^\dagger$ is iteratively updated with the improving $P_i^w$ during training, leading to self-evolving structural guidance with progressively reduced noise and increased reliability.

## 3.3. Semantic-spatial Prior with MLLM (SPM)

The reliability of pseudo-labels constitutes a core bottleneck in SS-RES, yet model predictions and MLLM often exhibit pronounced, sample-dependent discrepancies. To exploit their complementary strengths even under such conflicts, we propose SPM to produce more robust soft supervision and mitigate error propagation. SPM aims to obtain a reliable pixel-wise prior for unlabeled supervision by calibrating the external prior with the model's weak-view prediction. The key challenge is that $P_i^\dagger$ may be locally noisy, while $P_i^w$ can be over-confident in easy regions. We therefore fuse the two sources in logit space with a pixel-adaptive weight that accounts for both uncertainty and sample-level agreement. We quantify pixel-wise certainty with a simple margin-based score: $C(p) = (2p - 1)^2$, where $C(p) \in [0, 1]$ attains its minimum at $p = 0.5$, indicating maximal ambiguity. As $p$ moves away from $0.5$ and approaches $0$ or $1$, $C(p)$ increases monotonically and approaches $1$, corresponding to higher-confidence predictions. In addition, we measure sample-level reliability via an image-level agreement between the weak prediction and the external prior:

$$\mathcal{W}\left(P_i^w, P_i^\dagger\right) = \frac{1}{|\Omega|} \sum_{u \in \Omega} \left(1 - \left|P_i^w(u) - P_i^\dagger(u)\right|\right). \quad (4)$$

where $\Omega$ denotes the set of pixels and $u$ indexes pixel locations. $\mathcal{W}$ down-weights the prior when the two sources disagree globally. We then compute a pixel-wise fusion weight $\lambda(u) \in [0, 1]$:

$$\lambda(u) = \text{Clamp}_{[0,1]}\left(\lambda_0 + \kappa_p C(P_i^\dagger(u)) - \kappa_w C(P_i^w(u)) + \kappa_a\left(\mathcal{W}(P_i^w, P_i^\dagger) - 0.5\right)\right). \quad (5)$$

This design increases reliance on $P_i^\dagger$ when the model is uncertain, while suppressing the prior when the model is already confident or when the global agreement is low. Finally, we perform fusion in the logit space. Let $\ell(p) = \log\frac{p}{1-p}$ and $\sigma(\cdot)$ be the sigmoid. For numerical stability, we define $\hat{P} = \text{Clamp}_{[\epsilon, 1-\epsilon]}(P)$ and compute the calibrated prior:

$$\tilde{P}_i^\dagger(u) = \sigma\left(\lambda(u)\,\ell(\hat{P}_i^\dagger(u)) + (1 - \lambda(u))\,\ell(\hat{P}_i^w(u))\right). \quad (6)$$

The resulting $\tilde{P}_i^\dagger$ is used as calibrated soft supervision and as structural guidance for subsequent modules.

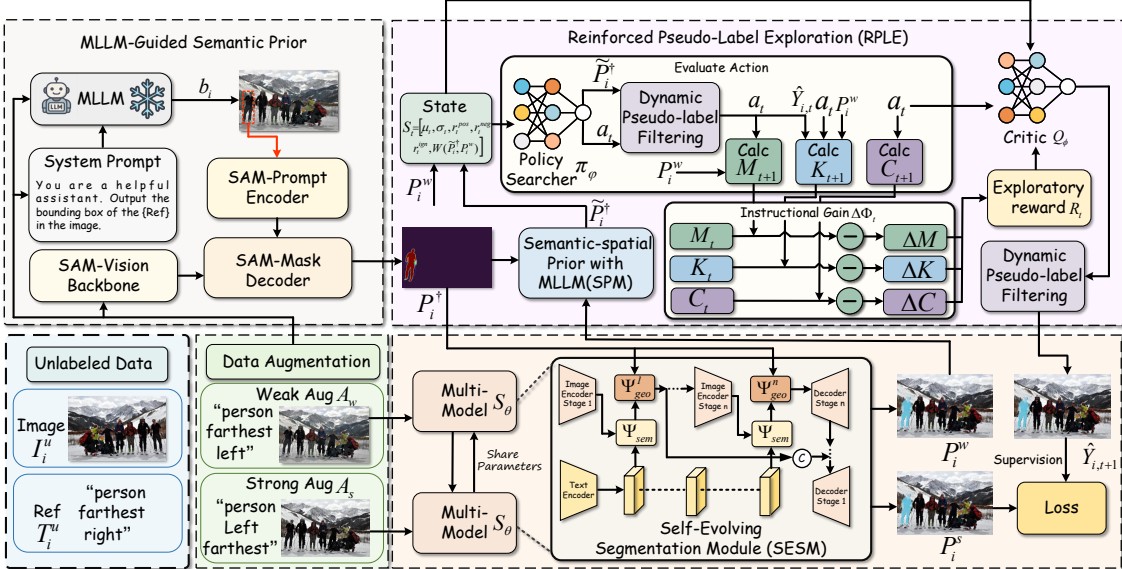

*Figure 3.* Overview of the proposed Learning to Label (L2L) framework. For each unlabeled image–ref pair, a frozen MLLM predicts referring grounding cues and prompts SAM2 to generate a soft segmentation prior, which is then uncertainty-adaptively calibrated by SPM using the model prediction under weak augmentation to obtain $\tilde{P}^\dagger$. The calibrated prior provides structured conditional guidance to the segmentor via SESM. Meanwhile, RPLE performs reinforced, sample-dependent pseudo-label filtering by predicting adaptive thresholds, enabling consistency-based self-training under strong augmentation. While MLLM-guided semantic priors are used during training, inference relies solely on the segmentation network $S_\theta$.

## 3.4. Reinforced Pseudo-Label Exploration (RPLE)

Since the optimal trade-off between noise suppression and coverage varies across training stages and sample difficulty, a fixed confidence threshold is often suboptimal. To this end, we formulate pseudo-label acceptance as a reinforced, sample-dependent exploration process. Policy Searcher $\pi_\varphi(s_t)$ dynamically produces thresholds, while Critic $Q_\phi(s, a)$ evaluates the selected pseudo-labels from multiple perspectives and provides feedback, enabling the learning of an adaptive pseudo-labeling policy. We define each reinforcement learning step $t$ as one self-training iteration on an unlabeled sample. $\pi_\varphi(s_t)$ perceives State $s_t \in \mathbb{R}^6$. To capture distributional characteristics and calibration status, we define:

$$s_t = [\mu_t, \sigma_t, r_t^{pos}, r_t^{neg}, r_t^{ign}, \mathcal{W}(P_i^w, \tilde{P}_i^\dagger)], \quad (7)$$

where $\mu_t$ and $\sigma_t$ denote the mean and standard deviation of the fused prediction $\tilde{P}_i^\dagger$, reflecting the global confidence distribution; $r_t^{pos}$, $r_t^{neg}$, and $r_t^{ign}$ are the proportions of pixels classified as foreground, background, and ignored under the current thresholds, respectively; and $\mathcal{W}(P_i^w, \tilde{P}_i^\dagger)$ is the sample-level agreement score, measuring global consistency between the MLLM prior and the model prediction. This state representation provides richer decision cues than a single confidence statistic.

Conditioned on $s_t$, the Actor outputs continuous adaptive thresholds $a_t = (\tau_t^{fg}, \tau_t^{bg}) \in [0, 1]$, representing the foreground and background confidence thresholds, respectively. We assign pixels with $\tilde{P}_i^\dagger(u) \geq \tau_t^{fg}$ as foreground, those

with $\tilde{P}_i^\dagger(u) \leq \tau_t^{bg}$ as background, and ignore the rest. This yields a binary pseudo-label map $\hat{Y}_{i,t}$ and the selected set $\Omega_t^{sel} = \Omega_t^{pos} \cup \Omega_t^{neg}$. The Critic $Q_\phi(s, a)$ serves as a multi-perspective quality evaluator, guiding the Actor to learn an adaptive thresholding policy. To train such a policy, the reward should reflect the utility of the selected pseudo-labels along complementary dimensions. Accordingly, we introduce a multi-perspective Instructional Gain $\Delta\Phi_t$ to evaluate the effect of action $a_t$: where $w_m$, $w_k$, and $w_c$ are weighting coefficients, and $\mathcal{P}_{stab}$ penalizes unstable threshold changes.

$$\Delta\Phi_t = \underbrace{w_m(\mathcal{M}_{t+1} - \mathcal{M}_t)}_{\text{Separability Gain } \Delta\mathcal{M}} + \underbrace{w_k(\mathcal{K}_{t+1} - \mathcal{K}_t)}_{\text{Consistency Gain } \Delta\mathcal{K}}$$
$$+ \underbrace{w_c(\mathcal{C}_{t+1} - \mathcal{C}_t)}_{\text{Coverage Gain } \Delta\mathcal{C}} - \mathcal{P}_{stab}, \quad (8)$$

The separability score $\mathcal{M}_t$ quantifies the contrastive boundary in the probability space induced by model predictions:

$$\mathcal{M}_t = \frac{1}{|\Omega_t^{sel}|} \sum_{u \in \Omega_t^{sel}} \sigma(\beta \cdot (\underbrace{\mathbf{v}_u^\top \mathbf{c}_{\hat{Y}_{i,t}(u),t}}_{\text{Intra-class}} - \underbrace{\mathbf{v}_u^\top \mathbf{c}_{1-\hat{Y}_{i,t}(u),t}}_{\text{Inter-class}})).$$
$$(9)$$

where $\sigma(\cdot)$ denotes the sigmoid function and $\beta$ is a temperature scalar. For each class $k \in 0, 1$, $\mathbf{c}_{k,t}$ is the $\ell_2$-normalized sum of $\mathbf{v}_u$ over pixels assigned to class $k$ at step $t$. Larger $\mathcal{M}_t$ indicates that the selected pixels yield stronger foreground–background separability.

The consistency score $\mathcal{K}_t$ measures the agreement between the model prediction and the calibrated prior over the se-

*Table 1.* **Comparison of SS-RES results on RefCOCO, RefCOCO+, and RefCOCOg datasets under different label budgets (i.e., 0.1%, 1%, 5%, and 10%).** Here Supervised denotes training with labeled data only, Baseline (w/o MLLM) denotes a semi-supervised baseline without using MLLM priors, Baseline (w/ MLLM) denotes a naive linear fusion baseline that linearly blends the baseline prediction and the MLLM prior with a fixed mixing coefficient of 0.5.

| Methods | RefCOCO | | | | | | | | | | | |
| --- | --- | --- | --- | --- | --- | --- | --- | --- | --- | --- | --- | --- |
| | 0.1% | | | 1% | | | 5% | | | 10% | | |
| | val | testA | testB | val | testA | testB | val | testA | testB | val | testA | testB |
| Supervised | 16.91 | 19.22 | 14.72 | 28.02 | 31.32 | 24.35 | 43.58 | 48.16 | 39.55 | 51.81 | 55.35 | 48.38 |
| Fixmatch (Sohn et al., 2020) | 18.82 | 21.05 | 16.74 | 33.89 | 38.46 | 30.17 | 49.18 | 52.81 | 44.30 | 54.55 | 57.91 | 51.92 |
| AugSeg (Zhao et al., 2023) | 4.50 | 5.79 | 2.19 | 24.42 | 26.23 | 22.18 | 30.99 | 34.81 | 27.93 | 39.63 | 32.45 | 32.00 |
| CCVC (Wang et al., 2023) | 15.22 | 16.32 | 15.16 | 28.14 | 32.23 | 24.29 | 46.62 | 49.92 | 42.37 | 53.13 | 56.48 | 49.51 |
| RESMatch (Zang et al., 2025) | 20.07 | 22.47 | 17.88 | 36.86 | 42.32 | 31.11 | 51.95 | 56.91 | 47.20 | 58.45 | 62.56 | 54.17 |
| SemiRES (Yang et al., 2024) | - | - | - | 50.90 | 57.54 | 44.48 | 61.31 | 66.64 | 55.94 | - | - | - |
| Baseline (w/o MLLM) | 21.72 | 24.42 | 23.86 | 44.69 | 51.51 | 37.42 | 58.23 | 64.24 | 52.46 | 62.20 | 67.02 | 56.98 |
| Baseline (w/ MLLM) | 62.25 | 68.02 | 55.44 | 62.99 | 68.41 | 57.91 | 64.07 | 69.65 | 58.37 | 65.01 | 69.85 | 61.17 |
| **Ours** | **65.08** | **68.53** | **59.39** | **65.85** | **68.93** | **61.34** | **67.30** | **71.46** | **62.60** | **67.92** | **71.53** | **62.63** |

| Methods | RefCOCO+ | | | | | | | | | | | |
| --- | --- | --- | --- | --- | --- | --- | --- | --- | --- | --- | --- | --- |
| | 0.1% | | | 1% | | | 5% | | | 10% | | |
| | val | testA | testB | val | testA | testB | val | testA | testB | val | testA | testB |
| Supervised | 17.44 | 19.28 | 15.73 | 25.62 | 28.53 | 22.56 | 36.05 | 39.87 | 30.71 | 40.49 | 44.48 | 34.50 |
| Fixmatch (Sohn et al., 2020) | 15.41 | 19.72 | 10.53 | 29.29 | 33.47 | 25.55 | 38.81 | 43.34 | 33.31 | 42.50 | 47.81 | 36.57 |
| AugSeg (Zhao et al., 2023) | 5.69 | 10.13 | 0.63 | 22.43 | 25.35 | 19.49 | 33.23 | 36.68 | 28.22 | 35.07 | 38.48 | 31.17 |
| CCVC (Wang et al., 2023) | 14.59 | 15.30 | 14.77 | 29.12 | 32.44 | 25.31 | 37.30 | 41.37 | 32.20 | 41.38 | 45.24 | 35.76 |
| RESMatch (Zang et al., 2025) | 18.11 | 23.54 | 15.37 | 31.09 | 35.72 | 26.07 | 39.97 | 44.66 | 33.56 | 45.03 | 51.22 | 37.97 |
| SemiRES (Yang et al., 2024) | - | - | - | 36.49 | 42.86 | 28.58 | 47.00 | 54.52 | 38.74 | - | - | - |
| Baseline (w/o MLLM) | 21.05 | 24.08 | 16.76 | 34.45 | 40.44 | 26.81 | 47.05 | 52.67 | 37.81 | 51.11 | 57.96 | 41.36 |
| Baseline (w/ MLLM) | 53.37 | 60.35 | 43.80 | 54.59 | 60.41 | 45.26 | 56.14 | 62.52 | 46.25 | 56.24 | 62.77 | 48.22 |
| **Ours** | **55.97** | **61.45** | **48.19** | **56.23** | **62.05** | **47.82** | **57.44** | **63.60** | **50.43** | **57.95** | **63.74** | **50.26** |

| Methods | RefCOCOg | | | | | | | |
| --- | --- | --- | --- | --- | --- | --- | --- | --- |
| | 0.1% | | 1% | | 5% | | 10% | |
| | val-u | test-u | val-u | test-u | val-u | test-u | val-u | test-u |
| Supervised | 16.97 | 17.14 | 26.23 | 26.92 | 34.58 | 36.50 | 40.43 | 41.58 |
| Fixmatch (Sohn et al., 2020) | 19.96 | 21.12 | 31.62 | 33.58 | 39.55 | 40.98 | 45.50 | 46.78 |
| AugSeg (Zhao et al., 2023) | 1.05 | 0.94 | 24.07 | 24.71 | 34.08 | 35.31 | 37.02 | 38.28 |
| CCVC (Wang et al., 2023) | 11.58 | 11.70 | 29.35 | 30.57 | 37.76 | 39.23 | 42.50 | 43.49 |
| RESMatch (Zang et al., 2025) | 20.97 | 20.91 | 32.18 | 33.21 | 39.77 | 41.42 | 45.24 | 47.39 |
| SemiRES (Yang et al., 2024) | - | - | 34.76 | 36.18 | 47.61 | 50.11 | - | - |
| Baseline (w/o MLLM) | 21.25 | 22.32 | 34.96 | 35.87 | 46.45 | 48.24 | 49.26 | 50.81 |
| Baseline (w/ MLLM) | 50.13 | 51.87 | 51.97 | 53.41 | 57.74 | 59.84 | 59.74 | 61.51 |
| **Ours** | **51.01** | **52.80** | **53.83** | **56.00** | **59.71** | **61.11** | **60.63** | **62.66** |

lected pixels:

$$\mathcal{K}_t = \frac{1}{|\Omega_t^{sel}|} \sum_{u \in \Omega_t^{sel}} \left( 1 - \left| P_i^w(u) - \tilde{P}_i^\dagger(u) \right| \right), \quad (10)$$

where larger values indicate stronger alignment, yielding more reliable learning signals. The coverage is defined as the fraction of selected pixels:

$$\mathcal{C}_t = r_t^{pos} + r_t^{neg} = \frac{|\Omega_t^{sel}|}{|\Omega|}. \quad (11)$$

Using the instructional gain $\Delta\Phi_t$, we define an instantaneous exploration reward. For training stability, we apply reward clipping:

$$R_t = \max(-\delta, \min(\delta, \Delta\Phi_t)), \quad (12)$$

where $\delta$ bounds the reward magnitude (e.g., $\delta = 0.5$). The target value is defined as the sum of $R_t$ and the discounted future return estimated by the target Critic $Q_{\phi'}$:

$$y_t = R_t + \gamma\, Q_{\phi'}(s_{t+1}, \pi_\varphi(s_{t+1})). \quad (13)$$

We update the pseudo-label mining critic $Q_\phi$ by minimizing the temporal-difference error:

$$\mathcal{L}_{critic} = \mathbb{E}\left[(y_t - Q_\phi(s_t, a_t))^2\right]. \quad (14)$$

Under this formulation, the policy searcher is optimized via deterministic policy gradients to maximize the expected cumulative quality score, thereby dynamically calibrating thresholds toward near-optimal values.

## 4. Experiment

### 4.1. Experimental Setup

**Datasets:** We evaluate our method on three widely adopted RES benchmarks, namely RefCOCO (Yu et al., 2016), RefCOCO+ (Yu et al., 2016), and RefCOCOg (G-Ref) (Mao et al., 2016; Nagaraja et al., 2016), all built upon MS-COCO images (Lin et al., 2014). RefCOCO expressions frequently involve absolute/relative spatial cues, RefCOCO+ places more emphasis on appearance attributes by discouraging location words, while RefCOCOg typically contains longer and semantically richer descriptions, making the grounding problem more challenging (Nagaraja et al., 2016; Mao et al., 2016). For semi-supervised learning, we split the training

*Table 2.* **Comparison with zero-shot methods.** Overall IoU on RefCOCO/RefCOCO+/RefCOCOg, the best results are highlighted in **bold**, and the second-best results are underlined.

| Method | RefCOCO | | | RefCOCO+ | | | RefCOCOg | |
|---|---|---|---|---|---|---|---|---|
| | val | testA | testB | val | testA | testB | val-u | test-u |
| GL-CLIP (Yu et al., 2023) | 32.9 | 34.9 | 30.1 | 38.4 | 42.1 | 32.7 | 42.0 | 42.0 |
| CaR (Sun et al., 2024) | 33.6 | 35.4 | 30.5 | 34.2 | 36.0 | 31.0 | 36.7 | 36.6 |
| Ref-Diff (Ni et al., 2023) | 37.2 | 38.4 | 37.2 | 37.3 | 40.5 | 33.0 | 44.0 | 44.5 |
| TAS (Suo et al., 2023) | 39.8 | 41.1 | 36.2 | 43.6 | 49.1 | 36.5 | 46.6 | 46.8 |
| IteRPrimeE (Wang et al., 2025c) | 40.2 | 46.5 | 33.9 | 44.2 | 51.6 | 35.3 | 46.0 | 45.1 |
| Pseudo-RIS (Yu et al., 2024) | 41.1 | 48.2 | 33.5 | 44.3 | 51.4 | 35.1 | 46.0 | 46.7 |
| LGD+DINO (Li et al., 2025) | 49.5 | 54.7 | 41.0 | 49.6 | 58.4 | 38.6 | 50.3 | 51.1 |
| VLM-VG (Wang et al., 2025a) | 49.9 | 53.1 | 46.7 | 42.7 | 47.3 | 36.2 | 48.0 | 48.5 |
| HybridGL (Liu & Li, 2025) | 49.5 | 53.4 | 45.2 | 43.4 | 49.1 | 37.2 | 51.3 | 51.6 |
| SAM 3 Agent (Carion et al., 2025) | 59.4 | 64.3 | 55.0 | 51.4 | 57.0 | 44.9 | 57.2 | 58.8 |
| **Ours** (0.1% labeled settings) | 65.08 | 68.53 | 59.39 | 55.97 | 61.45 | 48.19 | 51.01 | 52.80 |

*Table 3.* **Ablation Study of Core Components.** Core components are progressively integrated into the baseline, metrics reported are Overall IoU.

| Baseline | SPM | SESM | RPLE | val | testA | testB |
|---|---|---|---|---|---|---|
| ✓ | - | - | - | 58.23 | 64.24 | 52.46 |
| ✓ | ✓ | - | - | 65.73 | 69.66 | 61.03 |
| ✓ | ✓ | ✓ | - | 66.99 | 70.73 | 61.92 |
| ✓ | ✓ | - | ✓ | 66.71 | 70.39 | 61.66 |
| ✓ | ✓ | ✓ | ✓ | **67.30** | **71.46** | **62.60** |

set into a labeled subset and an unlabeled subset (e.g., 10% labeled + 90% unlabeled) under the same data distribution and report performance on the standard evaluation splits.

**Implementation Details:** Our model is implemented in PyTorch, adopting a Swin Transformer visual backbone (Liu et al., 2021) and a BERT text encoder (Devlin, 2018). Unless otherwise specified, we optimize the network using AdamW (Loshchilov & Hutter, 2019). All experiments are conducted for 40 epochs on 4 NVIDIA RTX 4090 GPUs with a batch size of 4. Images are resized to $480 \times 480$. Following the common weak-to-strong semi-supervised paradigm (Sohn et al., 2020; Zang et al., 2025), we employ weak and strong augmentations to construct consistency-based supervision on unlabeled data. Complete datasets and implementation details are provided in Appendix B.

### 4.2. Experimental Results

As shown in Table 1, we perform performance comparisons between our proposed L2L method and existing SS-RES methods on the RefCOCO, RefCOCO+, and RefCOCOg datasets under four label budgets. L2L achieves the best overall IoU across all settings. Compared to self-training that relies solely on model-generated pseudo-labels (w/o MLLM) and to the baseline (w/ MLLM) with a fixed mixing coefficient of 0.5, L2L is more stable across different budgets and shows a more pronounced advantage in low-label regimes. Figure 4 further presents qualitative visualizations. For the query "Boy holding pizza", L2L effectively suppresses distracting regions and recovers a more complete mask of the referred person. For "Donut on right with

*Table 4.* **Analysis of Weighting Strategies in SESM.** Comparison of single-stream, fixed-weight, and ours for integrating the semantic ($\alpha$) and structural ($\beta$) streams in SESM.

| Method | val | testA | testB |
|---|---|---|---|
| *Single-Stream Baselines* | | | |
| $\mathcal{S}$ ($\alpha$) | 66.33 | 70.23 | 62.54 |
| $\mathcal{G}_i$ ($\beta$) | 66.70 | 70.09 | 62.55 |
| *Multi-Stream Integration* | | | |
| Fixed Weighting | 67.04 | 70.74 | 62.15 |
| **Ours** | **67.30** | **71.46** | **62.60** |

*Table 5.* **Interaction Ablation on Stages.** Ablation study of the structural-stream configuration across encoder stages. $\mathcal{P}$ denotes the projection operator and $\mathcal{A}$ denotes the attention operator; configurations list the operator used at each stage from S1 to S4.

| S1 | S2 | S3 | S4 | val | testA | testB |
|---|---|---|---|---|---|---|
| $\mathcal{P}$ | $\mathcal{P}$ | $\mathcal{P}$ | $\mathcal{P}$ | 66.92 | 69.66 | 61.80 |
| $\mathcal{A}$ | $\mathcal{A}$ | $\mathcal{A}$ | $\mathcal{A}$ | 65.08 | 68.54 | 59.13 |
| $\mathcal{A}$ | $\mathcal{A}$ | $\mathcal{P}$ | $\mathcal{P}$ | 65.19 | 69.51 | 60.25 |
| $\mathcal{P}$ | $\mathcal{P}$ | $\mathcal{A}$ | $\mathcal{A}$ | **67.30** | **71.46** | **62.60** |

striped glaze", it successfully distinguishes the target donut from similar instances. For "Dog left", L2L remains robust to salient distractors and still generates an accurate mask for the referred object.

To provide a more stringent evaluation, we further report results under a near-zero-shot setting, positioning our method in an extremely low-label regime and benchmarking it against representative zero-shot baselines. Specifically, we adopt the most label-scarce configuration (0.1% labeled images): 16 labeled images on RefCOCO and RefCOCO+, and 21 labeled images on RefCOCOg, making the annotation cost comparable to a near-zero-shot scenario. Table 2 summarizes the comparison with representative zero-shot baselines. Notably, to keep the comparison with SAM 3 Agent (Carion et al., 2025) as controlled and fair as possible, we instantiate both methods with the same MLLM (Qwen2.5-VL 7B). Despite using only a handful of labeled samples, L2L yields substantial performance gains and surpasses zero-shot baselines on most splits, indicating that limited supervision can effectively leverage external priors.

### 4.3. Ablation Studies

**Effect of core components.** Table 3 summarizes the contribution of different components in L2L under the RefCOCO 5% labeled setting. Adding SPM on top of the semi-supervised baseline yields an improvement, suggesting that uncertainty-aware fusion can mitigate noisy guidance from foundation priors and provide more reliable training signals. Building upon SPM, incorporating SESM brings further gains, indicating that injecting semantic and structural cues into the encoder helps learn stronger representations under limited supervision. Moreover, introducing RPLE consis-

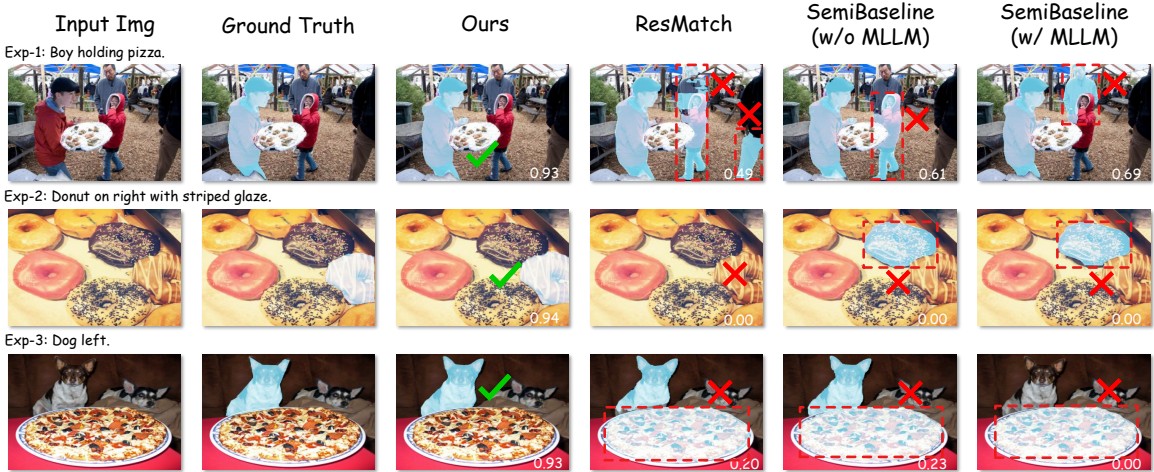

*Figure 4.* **Qualitative analysis of Ground Truth, L2L, RESMatch, Baseline (w/o MLLM), and Baseline (w/ MLLM) on RefCOCO under the 5% semi-supervised setting.** The white numbers indicate the IoU with the ground-truth mask. Typical failure regions are highlighted with red dashed boxes and incorrect results are marked in red; correct segmentations are marked in green.

*Table 6.* **Sensitivity to SPM coefficients.** The SPM weighting coefficients $(\kappa_p, \kappa_w, \kappa_a)$ are swept in a grid, and performance is evaluated on RefCOCO val/testA/testB. Results indicate that the method is insensitive to coefficient variations.

| $\kappa_p$ | $\kappa_w$ | $\kappa_a$ | val | testA | testB |
|------|------|------|------|-------|-------|
| 0.25 | 0.25 | 0.20 | 66.92 | 71.26 | **63.02** |
| 0.40 | 0.25 | 0.20 | **67.30** | **71.46** | 62.60 |
| 0.40 | 0.40 | 0.20 | 66.45 | 70.21 | 62.33 |
| 0.40 | 0.40 | 0.40 | 65.96 | 69.82 | 60.89 |
| 0.40 | 0.40 | 0.60 | 66.61 | 71.04 | 61.78 |

*Table 7.* Reward-component ablation for RPLE under the Ref-COCO 5% labeled setting.

| Method | val | testA | testB |
|--------|------|-------|-------|
| w/o $\Delta M$ | 66.35 | 70.21 | 62.54 |
| w/o $\Delta K$ | 65.36 | 69.12 | 60.88 |
| w/o $\Delta C$ | 65.95 | 69.64 | 61.56 |
| Ours | **67.30** | **71.46** | **62.60** |

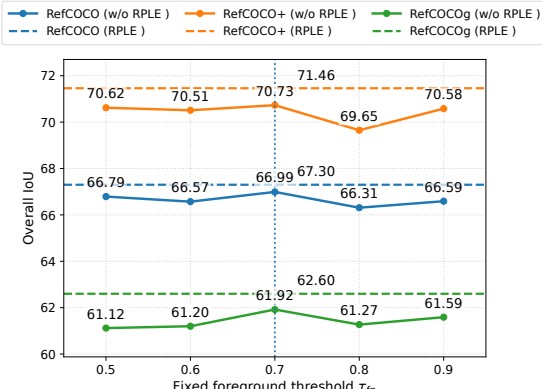

*Figure 5.* **Sensitivity to a fixed foreground threshold.** Overall IoU on RefCOCO/RefCOCO+/RefCOCOg under the 5% labeled setting when replacing RPLE with a static threshold $\tau_{fg}$. RPLE results are shown as dashed horizontal lines, achieving comparable or better performance without manual threshold tuning. The vertical dotted line marks the default fixed threshold $\tau_{fg} = 0.7$.

tently improves performance, implying that a policy-driven, adaptive pseudo-label selection strategy can better control noise propagation and enhance training robustness.

**Weighting strategies in SESM.** Table 4 investigates how to combine the semantic stream and the structural stream in SESM under the RefCOCO 5% labeled setting. The results show that using a single stream generally yields inferior performance, while integrating both streams further improves accuracy. In particular, fixed weighting fuses two streams with equal weights, i.e., $\alpha = \beta = 0.5$, and outperforms single-stream variants, SESM achieves the best results. These findings support our motivation that the relative importance of semantic and structural cues varies across samples, and thus should be adjusted dynamically.

**Stage-wise structural interaction in SESM.** Table 5 compares different structural-stream configurations across encoder stages under the RefCOCO 5% labeled setting. We find that a hybrid design performs best, emphasizing local alignment in shallow stages (S1–S2) and conducting relational reasoning in deeper stages (S3–S4). This is consistent with the stage-wise nature of hierarchical vision backbones, where early layers capture spatial details while deeper layers encode higher-level semantics.

**Sensitivity to SPM coefficients.** We compare several settings of the SPM coefficients, and report the results in Table 6. Overall, the performance gap across different settings is small, indicating that SPM does not require delicate tuning. Meanwhile, we observe a consistent trend: increasing $\kappa_w$ or $\kappa_a$ is more likely to degrade performance, whereas varying $\kappa_p$ has a relatively minor effect.

**Ablation on RPLE reward components.** We further study the contribution of each reward component in RPLE under the RefCOCO 5% labeled setting. As shown in Table 7, removing any component from the exploratory reward degrades the performance, indicating that separability gain $\Delta M$, consistency gain $\Delta K$, and coverage gain $\Delta C$ provide complementary supervision for pseudo-label exploration. Among them, removing $\Delta K$ leads to the largest drop, suggesting that maintaining agreement between the calibrated prior and the model prediction is particularly important for reliable pseudo-label selection.

**RPLE vs. Static Thresholding for Pseudo-label Selection.** To evaluate the effect of RPLE on pixel-level pseudo-label selection, we replace RPLE with a static foreground threshold while keeping all other training settings of L2L unchanged, and sweep $\tau_{fg} \in \{0.5, 0.6, 0.7, 0.8, 0.9\}$. As shown in Figure 5, static thresholding is highly sensitive to the choice of $\tau_{fg}$, with performance varying noticeably across different thresholds, and thus requires careful tuning. In contrast, RPLE achieves comparable or even better results without manual threshold selection.

## 5. Conclusion

We propose Learning to Label (L2L), a reinforced self-evolving framework for semi-supervised referring expression segmentation. L2L integrates MLLM semantic–spatial priors with uncertainty-aware fusion to provide reliable guidance, and introduces a reinforcement-driven, sample-adaptive pseudo-label strategy to progressively improve supervision on unlabeled data. Extensive experiments under label budgets ranging from 0.1% to 10% demonstrate consistent improvements across settings; notably, L2L maintains strong performance and robustness even with limited labeled samples. We believe that this work provides insights for building reliable, data-efficient, and generalizable referring segmentation systems under extremely low annotation costs.

## Acknowledgement

This work was supported by the National Natural Science Foundation of China (Grant Nos. 62372238 and 62476133), the National Science and Technology Major Project of China (Grant No. 2024ZD0524600), the Fundamental Research Funds for the Central Universities (Grant No. 11300-312200502507), the National Science Foundation of China (Grant No. 625B2169), ZJU Kunpeng & Ascend Center of Excellence, and Dream Set Off - Kunpeng & Ascend Seed Program, the National Natural Science Foundation of China under Grants 62402245, the Natural Science Foundation of Jiangsu Province under Grants BK20240644, and the Natural Science Foundation of the Jiangsu Higher Education Institutions of China under Grants 24KJB520023, and the Youth Talent Support Program of the Jiangsu Association for Science and Technology (Grant No. JSTJ-2025-952).

## Impact Statement

This paper presents work whose goal is to advance the field of Machine Learning. There are many potential societal consequences of our work, none of which we feel must be specifically highlighted here.

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

# Appendix of Learning to Label: A Reinforced Self-Evolving Framework for Semi-supervised Referring Expression Segmentation

In the appendix, we include additional content to complement the main paper, including further discussions and extended experimental results.

- **Appendix A**: Additional method details and extensions (cf. section 3 in the main paper).

- **Appendix B**: Full experimental settings and additional results (cf. section 4 in the main paper).

## A. More Details about Method

### A.1. Semi-Supervised Baseline

Following standard semi-supervised learning practices (Sohn et al., 2020; Zang et al., 2025), we adopt a weak-to-strong consistency regularization baseline for SS-RES. We are given a labeled set $\mathcal{D}_l = \{(I_i^l, T_i^l, Y_i^l)\}_{i=1}^{N_l}$ and an unlabeled set $\mathcal{D}_u = \{(I_i^u, T_i^u)\}_{i=1}^{N_u}$, where $I$, $T$, and $Y$ denote the image, referring expression, and pixel-wise binary mask, respectively. Unless stated otherwise, we use $P \in [0,1]^{H \times W}$ to denote a foreground probability map.

For labeled data, the segmentation network $S_\theta$ predicts $P_i^l = S_\theta(I_i^l, T_i^l)$ and is optimized with the pixel-wise binary cross-entropy (BCE):

$$\mathcal{L}_{\text{sup}} = \frac{1}{N_l} \sum_{i=1}^{N_l} \mathcal{L}_{\text{bce}}(P_i^l, Y_i^l). \tag{15}$$

We adopt the standard pixel-wise BCE,

$$\mathcal{L}_{\text{bce}}(P, Y) = -\frac{1}{|\Omega|} \sum_{u \in \Omega} \Big( Y(u) \log P(u) + (1 - Y(u)) \log(1 - P(u)) \Big), \tag{16}$$

where $\Omega$ indexes pixel locations. For each unlabeled pair $(I_i^u, T_i^u) \in \mathcal{D}_u$, we obtain weak/strong predictions using weak and strong augmentations:

$$P_i^w = S_\theta(\mathcal{A}_w(I_i^u, T_i^u)), \qquad P_i^s = S_\theta(A_s(A_w(I^u, T^u))). \tag{17}$$

A static confidence threshold $\tau$ is used to construct pseudo supervision from the weak prediction. We binarize $P_i^w$ element-wise to form a pseudo-mask $\hat{Y}_i^u = \mathbb{I}[P_i^w \geq 0.7]$, and define a confidence mask $M_i = \mathbb{I}[P_i^w \geq \tau]$ to select reliable pixels. The unsupervised loss is computed on the strong prediction over selected pixels:

$$\mathcal{L}_{\text{unsup}} = \frac{1}{N_u} \sum_{i=1}^{N_u} \frac{1}{\sum_{u \in \Omega} M_i(u) + \epsilon} \sum_{u \in \Omega} M_i(u) \cdot \mathcal{L}_{\text{bce}}\big(P_i^s(u), \hat{Y}_i^u(u)\big), \tag{18}$$

where $\mathbb{I}[\cdot]$ is the indicator function and $\epsilon$ is a small constant to avoid numerical issues when no pixel is selected. The pseudo-mask $\hat{Y}_i^u$ is treated as a fixed target (i.e., gradients are not back-propagated through $\hat{Y}_i^u$). When $\mathcal{A}_s$ includes geometric transformations, pseudo supervision is applied in the aligned coordinate system induced by the corresponding augmentations. The final training objective is

$$\mathcal{L}_{\text{total}} = \mathcal{L}_{\text{sup}} + \lambda_u \mathcal{L}_{\text{unsup}}, \tag{19}$$

where $\lambda_u$ controls the contribution of unlabeled data. While this baseline provides a useful reference, a fixed threshold $\tau$ cannot accommodate the sample-specific ambiguity in referring segmentation, which may induce confirmation bias on easy samples and insufficient supervision on hard samples.

*Table 8.* **Image augmentations in our weak-to-strong framework.**

| Operation | Description and rationale | Range |
|---|---|---|
| **Weak augmentations** | | |
| Resize | Resize to a fixed resolution to obtain a stable view for pseudo-labeling. | – |
| RandomHorizontalFlip | Flip the image; for RES we also swap directional words in the expression (e.g., left↔right) to keep text–image semantics aligned. | 0.5 |
| **Strong augmentations** | | |
| Identity | No operation; included as an option in the augmentation pool. | – |
| AutoContrast | Automatic contrast normalization to perturb global appearance statistics. | – |
| Histogram | Histogram equalization to change intensity distribution without altering geometry. | – |
| GaussianBlur | Apply Gaussian blur to reduce high-frequency details. | $\sigma \in [0.1, 2.0]$ |
| Contrast | Adjust contrast while keeping spatial layout unchanged. | $\alpha \in [0.1, 0.95]$ |
| Sharpness | Adjust sharpness without changing geometry. | $\alpha \in [0.1, 0.95]$ |
| Color balance | Adjust color balance to perturb color statistics while preserving layout. | $\alpha \in [0.1, 0.95]$ |
| Brightness | Adjust brightness to simulate illumination variations. | $\alpha \in [0.1, 0.95]$ |
| Posterize | Reduce color bit-depth to induce quantization artifacts. | bits $\in [4, 8]$ |

## A.2. Semi-Supervised Data Augmentation

We follow the weak-to-strong consistency pseudo-labeling paradigm of FixMatch and adopt the task-adapted multimodal augmentation strategy introduced in RESMatch for RES (Sohn et al., 2020; Zang et al., 2025). Concretely, for each unlabeled image–text pair $(I^u, T^u)$, we first construct a weak view using the weak augmentation operator $A_w(\cdot)$, and then construct a strong view by applying the strong augmentation operator $A_s(\cdot)$ on top of the weak view, i.e., $A_s(A_w(I^u, T^u))$. We use the same segmentation network $S_\theta$ to predict on both views. Specifically, we obtain the prediction on the weak view

$$P_w = S_\theta\big(A_w(I^u, T^u)\big), \tag{20}$$

which is used to produce pseudo-label supervision. We also obtain the prediction on the strong view

$$P_s = S_\theta\big(A_s(A_w(I^u, T^u))\big). \tag{21}$$

We optimize $S_\theta$ by enforcing a consistency objective between $P_w$ and $P_s$. On the image side, the weak image augmentation consists of resizing and random horizontal flipping, which yields a relatively stable weak view. The strong view is constructed by sampling strong image augmentations from an intensity-based operator pool, as summarized in Table 8. These operators primarily perturb appearance statistics while largely preserving spatial layout and geometry, which is desirable in RES where alignment between spatial relations and referred regions is critical. For geometric transforms that change directional semantics (e.g., horizontal flip), we apply correlated text edits by swapping directional words in the expression (e.g., left↔right) to maintain language–image semantic alignment.

On the text side, in addition to the semantic-preserving edits correlated with geometric transforms, we employ EDA-style strong text augmentation to improve linguistic diversity and robustness, including synonym replacement, synonym insertion, random swap, and random deletion are shown in Table 9. In our implementation, we set $\alpha_{sr} = \alpha_{ri} = \alpha_{rs} = 0.1$ and $p_{rd} = 0.1$, and generate 9 augmented expressions per original expression; together with the original sentence, this results in 10 text candidates per sample. To mitigate semantic drift introduced by strong perturbations, we further apply semantic relevance filtering to these candidates: we compute the cosine similarity between BERT embeddings of the weak and augmented texts, discard candidates with insufficient semantic relevance, and avoid a greedy strategy that keeps only the most similar candidates so as to retain sufficient perturbation diversity.

To further illustrate what the strong augmentations change in both modalities, we provide qualitative examples in Figure 6. The figure shows that strong image augmentations induce substantial appearance variations while preserving the overall layout and target structure, and that strong text augmentations yield diverse paraphrases which remain highly consistent with the original expression after semantic filtering. This visualization helps clarify how we construct strong views and supports that our strong augmentation design introduces sufficient perturbations without materially breaking the referring semantic alignment, thereby providing effective training signals for consistency learning.

*Table 9.* **Text augmentations in our weak-to-strong framework.**

| Operation | Description and rationale | Parameter |
|---|---|---|
| **Weak augmentations** | | |
| Semantic-preserving change | Swap directional words to match geometric transforms (e.g., when flipped, left→right) to keep language–image alignment. | rule-based |
| **Strong augmentations** | | |
| Synonym Replacement | Replace a subset of non-stop words with synonyms to vary surface form while preserving meaning. | $\alpha_{\mathrm{sr}} = 0.1$ |
| Synonym Insertion | Insert synonyms of randomly selected non-stop words to increase expression diversity. | $\alpha_{\mathrm{ri}} = 0.1$ |
| Random Swap | Swap word positions to improve robustness to word-order noise. | $\alpha_{\mathrm{rs}} = 0.1$ |
| Random Deletion | Delete each word with a fixed probability to simulate incomplete/noisy expressions. | $p_{\mathrm{rd}} = 0.1$ |

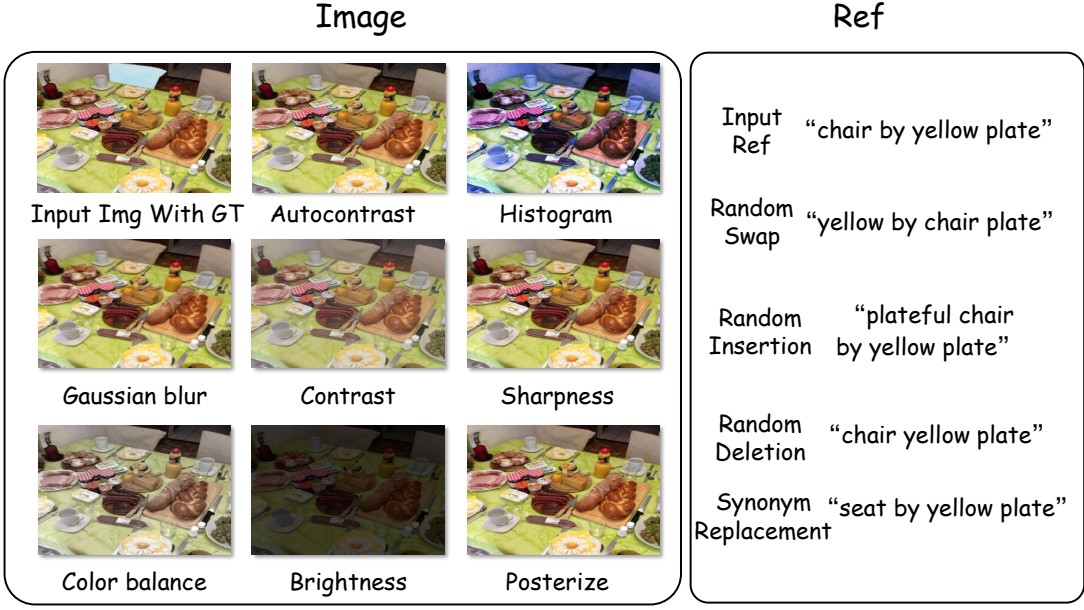

*Figure 6.* **Qualitative visualization of strong augmentations.** We show representative examples of strong image augmentations and strong text augmentations applied to the same unlabeled sample.

# B. More Details about Experiments

## B.1. Datasets Detail

**RefCOCO.** RefCOCO (Yu et al., 2016) is a widely used RES benchmark built on MS-COCO images (Lin et al., 2014). Each instance is an image–expression pair with a binary mask as supervision. The benchmark contains 19,994 images, 50,000 annotated objects, and 142,209 referring expressions. We follow the official split with *train*, *val*, *testA*, and *testB*. The *testA* split is dominated by person instances while *testB* focuses on non-person categories, which is commonly used to analyze category-wise generalization. RefCOCO expressions are typically short and frequently include spatial cues, making it suitable for evaluating spatial grounding.

**RefCOCO+.** RefCOCO+ (Yu et al., 2016) shares the same image source as RefCOCO but discourages explicit location words, encouraging models to rely more on appearance attributes and contextual evidence. It contains 19,992 images, 49,856 objects, and 141,564 expressions. We use the same official *train/val/testA/testB* protocol. Compared to RefCOCO, RefCOCO+ tends to be more ambiguous linguistically because many expressions describe attributes rather than positions.

**RefCOCOg.** RefCOCOg (Mao et al., 2016) is also known as G-Ref and provides longer descriptions with richer compositional semantics and relational structures. The dataset includes 26,711 images, 54,822 objects, and 104,560 expressions. Following common practice, we adopt the UMD split and report results on *val-u* and *test-u*, with 23,199 training samples, 2,601 validation samples, and 4,010 test samples.

## B.2. Semi-supervised setting

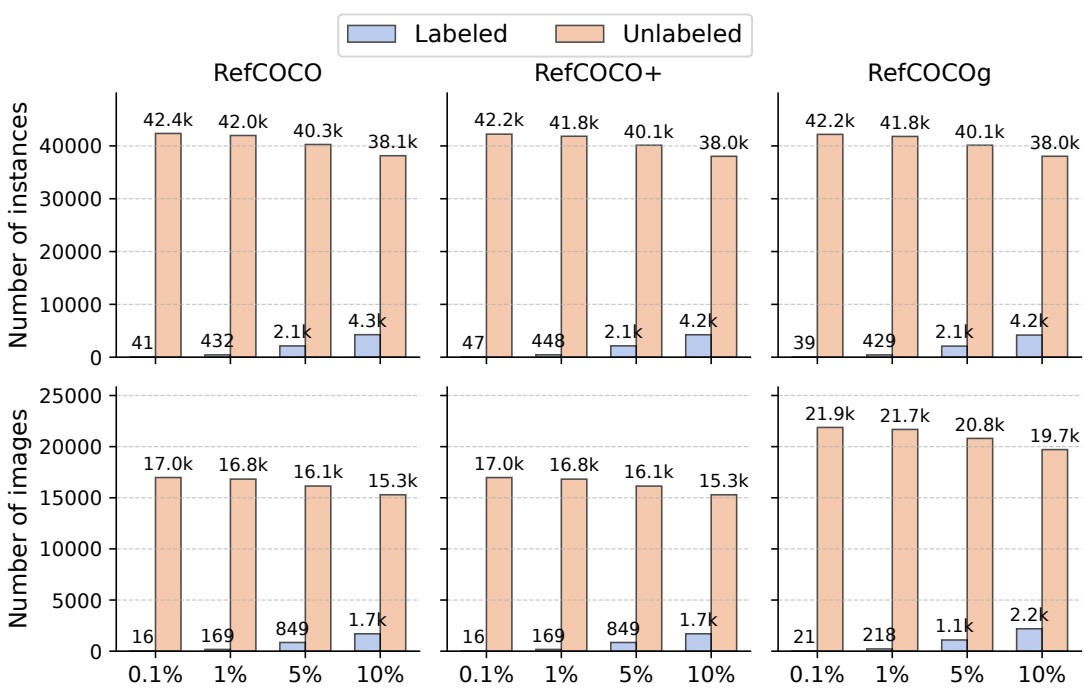

*Figure 7.* Labeled and unlabeled data statistics under four label budgets on RefCOCO, RefCOCO+ and RefCOCOg.

Following the semi-supervised RES protocol in (Sun et al., 2023; Yang et al., 2024; Zang et al., 2025), we study the in-distribution setting where the official training split is partitioned into a labeled subset and an unlabeled subset drawn from the same data distribution. We consider four label budgets, namely 0.1%, 1%, 5%, and 10%, and treat the remaining training samples as unlabeled. To avoid supervision leakage across subsets, we construct the labeled and unlabeled partitions at the image level. Figure 7 summarizes the data statistics of the labeled and unlabeled splits for RefCOCO, RefCOCO+, and RefCOCOg under different label budgets.

## B.3. Implementation Details

We implement our method in PyTorch and train it with distributed data parallelism. Unless otherwise specified, all experiments are conducted on 4 NVIDIA RTX 4090 GPUs for 40 epochs with a batch size of 4, and input images are resized to $480 \times 480$. We optimize the network using AdamW with a base learning rate of $5 \times 10^{-5}$ and weight decay of $10^{-2}$, and apply a polynomial learning-rate decay with power 0.9 over the whole training process. We fix the random seed to 22 for reproducibility.

**Supervised and unsupervised losses.** For labeled data, we adopt pixel-wise cross-entropy with ignore index 255 for invalid pixels, and use class weights $w_{bg} = 0.5$ and $w_{fg} = 1.5$ to alleviate foreground–background imbalance. For unlabeled data, our RPLE predicts image-specific foreground and background thresholds, denoted by $\tau_{fg}$ and $\tau_{bg}$, which partition pixels into confident foreground, confident background, and ignored regions. We apply cross-entropy on confident (non-ignored) pixels and add a Dice regularizer on confident foreground pixels. The overall objective is

$$\mathcal{L} = \lambda_l \, \mathcal{L}_l + \lambda_u \Big( \alpha \, \mathcal{L}_{bce} + (1 - \alpha) \, \mathcal{L}_{dice} \Big), \tag{22}$$

where $\mathcal{L}_l$ is the supervised term on labeled samples, and $\mathcal{L}_{bce}$ and $\mathcal{L}_{dice}$ are computed on unlabeled samples restricted to RPLE-selected confident regions. Unless stated otherwise, we set $\lambda_l = \lambda_u = 1.0$ and $\alpha = 0.7$.

**Semantic-spatial Prior with MLLM (SPM).** In implementation, SPM fuses the external prior (obtained by applying a foundation segmentor prompted by MLLM-generated grounding cues) with the model prediction on the weakly augmented view, producing a calibrated soft target for unlabeled supervision. The fusion weight is determined by pixel-wise confidence cues and an image-level agreement signal between the two sources, such that the external prior is down-weighted when it substantially conflicts with the model prediction or is deemed unreliable. Unless stated otherwise, the fusion coefficients are kept fixed throughout training and set to $\lambda_0 = 0.25$, $\kappa_p = 0.40$, $\kappa_w = 0.25$, and $\kappa_a = 0.20$ in Eq. 5.

**Reinforced Pseudo-Label Exploration (RPLE).** RPLE is initialized with $\tau_{fg} = 0.70$ and $\tau_{bg} = 0.20$. When constructing the foreground/background/ignored regions, we suppress ambiguous boundary supervision by marking a 3-pixel-wide band around region boundaries as ignored. Both the actor and critic are implemented as two-layer MLPs with hidden size 64 and trained using DDPG. We use learning rates of $10^{-4}$ for both the actor and critic, a discount factor of 0.9, and a soft-update coefficient of 0.005. During training, we add Gaussian exploration noise with standard deviation 0.01 when sampling actions. The reward combines separability gain, semantic-consistency gain, and coverage gain, weighted by $w_m = 1.0$, $w_k = 0.3$, and $w_c = 0.2$, respectively, together with a stability penalty weighted by $w_{\text{stab}} = 0.05$. The reward is clipped to $[-0.5, 0.5]$ for stable training.

## B.4. More Experiment Results

**Expression-length stratified evaluation.** To further examine how linguistic complexity affects pseudo-labeling, we stratify RefCOCO expressions into four length buckets: short (1–3 words), medium (4–6 words), long (7–10 words), and extra-long (10+ words), and report Overall IoU under the 5% labeled setting. As shown in Table 10, performance drops noticeably from short to longer expressions.

*Table 10.* **Expression-length stratified evaluation on 5% labeled RefCOCO setting.** Overall IoU on val/testA/testB grouped by expression length.

| Expression Length | val | testA | testB |
|---|---|---|---|
| Short | 69.74 | 73.34 | 65.55 |
| Medium | 63.60 | 68.37 | 59.09 |
| Long | 53.20 | 59.89 | 48.12 |
| Extra-long | 44.10 | 48.94 | 43.59 |

**Foundation-model sensitivity.** To assess robustness to foundation-model choices, the foundation segmentor (SAM vs. SAM2) and the frozen MLLM backbone (Qwen2.5-VL 3B vs. 7B) are varied while keeping the remaining components and the training protocol fixed. As shown in Table 11, these substitutions lead to only minor changes in performance, indicating that the proposed method is largely insensitive to the specific foundation-model instantiation and thus exhibits strong robustness.

*Table 11.* **Foundation-model sensitivity under the RefCOCO 5% labeled setting**. Varying the frozen MLLM backbone and the foundation segmentor with all other components and training settings fixed.

| MLLM backbone | Segmentor | val | testA | testB |
|---|---|---|---|---|
| Qwen2.5-VL-7B | SAM2 | **67.30** | **71.46** | 62.60 |
| Qwen2.5-VL-7B | SAM | 66.77 | 71.23 | 61.23 |
| Qwen2.5-VL-3B | SAM2 | 66.64 | 71.09 | **62.96** |

**Parameter comparison and trainable overhead.** As shown in Table 12, our method incurs only a modest increase in the number of *trainable* parameters relative to the Baseline, indicating that the observed improvements are not merely due to scaling up the learnable segmentation network. Specifically, the Baseline contains 227.740M trainable parameters, whereas our model contains 237.093M trainable parameters (+9.353M). This additional parameter budget primarily comes from the extra learnable components introduced into the segmentation network, which are designed to better fuse and exploit the priors and to support the subsequent self-evolving learning procedure. In contrast, the overall parameter count is dominated by the offline foundation models used to generate semantic–spatial priors (MLLM and SAM2). These models remain frozen during use and are employed only for offline prior generation; therefore, they do not increase the trainable parameter budget or the backpropagation overhead in end-to-end training.

*Table 12.* **Parameter scale comparison.** Trainable Params counts the parameters of the segmentation network optimized in end-to-end training. Total Params further includes the frozen foundation models used for offline semantic–spatial prior generation.

| Method | Trainable Params (M) | Total Params (M) |
|---|---|---|
| Baseline (w/o MLLM) | 227.740 | 227.740 |
| Baseline (w/ MLLM) | 227.740 | 8744.337 |
| Ours | 237.093 | 8753.690 |

**Sensitivity analysis of re-partitioning under the semi-supervised setting.** In the 5% semi-supervised setting on RefCOCO, to examine the sensitivity of our method to random data partitioning, we keep all other training configurations unchanged and perform two independent random re-samplings of the labeled and unlabeled subsets from the training set under the same labeled-ratio constraint, followed by re-training the model for each partition. As shown in Table 13, the performance variations across different random partitions are small, indicating that our method is not sensitive to the specific construction of the labeled set and exhibits good stability.

*Table 13.* Results on RefCOCO under the 5% semi-supervised setting with different random labeled splits.

| Random partition | val | testA | testB |
|---|---|---|---|
| Split #1 | 67.30 | 71.46 | 62.60 |
| Split #2 | 67.15 | 70.72 | 62.65 |
| Split #3 | 67.09 | 70.85 | 61.70 |

**Confidence Map Visualization.** Figure 8 presents pixel-wise prediction confidence maps for our method and its ablations under the 5% semi-supervised setting on RefCOCO. The confidence is visualized in grayscale, with higher intensity corresponding to higher confidence. Across diverse referring expressions and challenging scenes, the full model produces high-confidence regions that are more spatially concentrated on the referred target and exhibit better alignment with the ground-truth mask. In contrast, the baseline and ablated variants tend to show more diffuse confidence distributions or increased leakage to non-target regions, which is often associated with missed target parts and fragmented predictions. Moreover, removing SESM typically degrades spatial coherence and weakens confidence around object boundaries, whereas removing RPLE may lead to less stable confidence in ambiguous regions. These observations suggest that the proposed components contribute positively to improving spatial focus and confidence calibration.

**Analysis of Stronger MLLM-based Baselines** To address the concern that the improvement may mainly come from the external MLLM+SAM2 prior, we add stronger MLLM-based baselines under the RefCOCO 5% labeled setting. Besides the

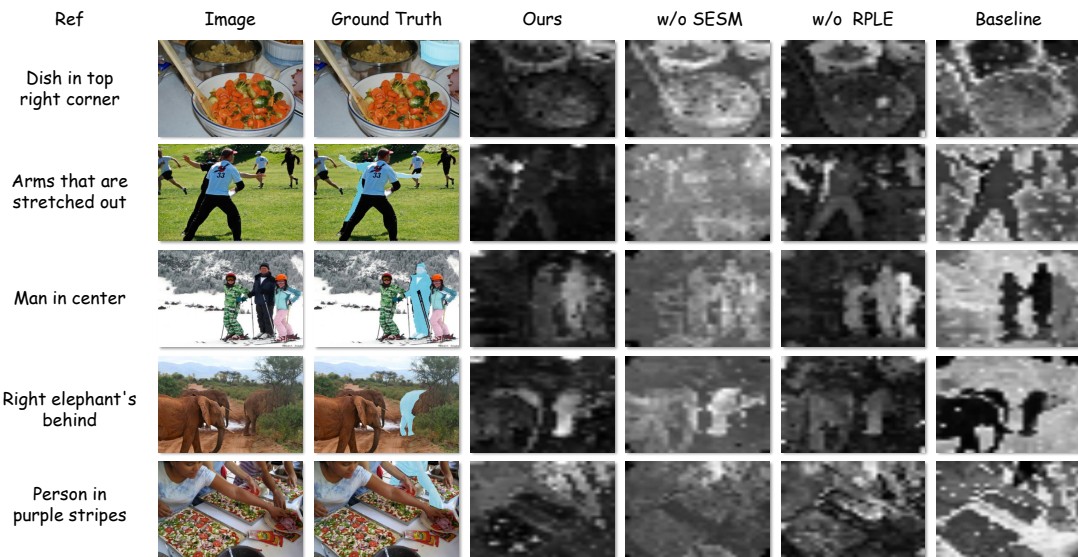

*Figure 8.* **Confidence map visualization** on RefCOCO under the 5% semi-supervised setting. From left to right: referring expression (Ref), input image (Image), ground-truth mask (Ground Truth), full model (Ours), ablation without SESM (w/o SESM), ablation without RPLE (w/o RPLE), and the baseline. Confidence is rendered in grayscale, where brighter pixels indicate higher confidence. Compared with the baseline and ablated variants, the full model yields more compact and spatially coherent high-confidence regions on the target, while reducing confidence leakage to irrelevant areas.

default Baseline (w/ MLLM), which uses a fixed linear fusion weight of 0.5, we further evaluate three stronger variants: tuned linear fusion, agreement-based filtering, and CATM-style adaptive thresholding (Moon et al., 2024). For tuned linear fusion, $\alpha$ and $\beta$ denote the weights of the baseline prediction and the MLLM prior, respectively, with $\alpha + \beta = 1$. As shown in Table 14, tuning the fusion weights brings moderate improvements over the fixed-fusion baseline, but the gain remains limited. For agreement-based filtering, we only retain pseudo-labels when the baseline prediction and the MLLM prior are both confident and their agreement is higher than 0.7, which is a stronger heuristic than naive fusion. We also compare with CATM-style adaptive thresholding (Moon et al., 2024), which replaces the fixed threshold with an adaptive pseudo-label selection strategy. As reported in Table 15, L2L still consistently outperforms these stronger baselines, indicating that the proposed modules exploit the external prior more effectively rather than merely benefiting from its existence.

*Table 14.* Effect of tuned mixing coefficients on the MLLM-enhanced baseline under the RefCOCO 5% labeled setting.

| $\alpha$ | $\beta = 1 - \alpha$ | **val** | **testA** | **testB** |
|---|---|---|---|---|
| 0.1 | 0.9 | 60.96 | 67.07 | 55.15 |
| 0.3 | 0.7 | 64.55 | 68.97 | **59.71** |
| 0.5 | 0.5 | 64.07 | **69.65** | 58.37 |
| 0.7 | 0.3 | 64.49 | 68.27 | 58.72 |
| 0.9 | 0.1 | **64.59** | 67.95 | 58.97 |

*Table 15.* Performance comparison with additional stronger MLLM-based baselines under the RefCOCO 5% labeled setting.

| Method | val | testA | testB |
|---|---|---|---|
| Baseline + Agreement filtering | 64.30 | 68.98 | 58.59 |
| Baseline + CATM | 65.05 | 69.50 | 59.90 |
| Ours | **67.30** | **71.46** | **62.60** |

**Computational Overhead Analysis** We further analyze the additional computational overhead introduced by L2L. The main extra cost comes from one-time offline prior generation rather than end-to-end optimization of foundation models during training. Specifically, the MLLM-based grounding cues and SAM-style mask priors are generated once for the unlabeled set and kept frozen throughout training. They are not updated by backpropagation. Table 16 reports the average

offline prior-generation time per image–expression pair on an NVIDIA RTX A6000 GPU. The reinforced pseudo-label exploration branch is lightweight. It is implemented as an actor–critic module with a 6-dimensional state and a 2-dimensional action space. The actor and critic contain only 0.009539M trainable parameters in total, or 0.019078M parameters when the target networks are included. The policy is updated once for each unlabeled mini-batch during training and is removed at inference time. Therefore, the additional overhead of RPLE is negligible compared with the segmentation network and the frozen foundation models, and L2L does not introduce extra inference-time cost.

*Table 16.* Average offline prior-generation time per image–expression pair.

| Metric | Qwen2.5-VL-7B | SAM1 | SAM2 |
|---|---|---|---|
| Avg. time (s/pair) | 1.7430 | 0.4541 | 0.1661 |

**Robustness to Non-MLLM Priors.** To examine whether L2L is restricted to MLLM-generated priors, we replace the original MLLM-based prior with priors produced by conventional non-MLLM referring segmentation models, while keeping the rest of L2L unchanged. Specifically, we use CARIS (Liu et al., 2023c) and CGFormer (Tang et al., 2023) to generate the initial external prior $\tilde{P}_i^{\dagger}$, and then apply L2L in the same way as the original setting. As shown in Table 17, L2L remains effective when the MLLM-based prior is replaced by non-MLLM teachers. Both CARIS and CGFormer priors achieve competitive performance, and the CARIS prior even obtains a higher result on testB. These results suggest that L2L is not tied to a specific MLLM teacher, but provides a general mechanism for calibrating and exploiting external priors.

*Table 17.* L2L with non-MLLM teachers on RefCOCO under the 5% labeled setting.

| Method | val | testA | testB |
|---|---|---|---|
| w/ CARIS prior | 67.04 | 70.80 | **64.02** |
| w/ CGFormer prior | 66.46 | 70.42 | 62.82 |
| Ours | **67.30** | **71.46** | 62.60 |

**Temporal Evolution of Pseudo Supervision Quality.** We further track the quality of pseudo supervision over the 40-epoch training process. As shown in Figure 9, the original MLLM-guided prior $P_i^{\dagger}$ remains fixed at 68.48 IoU, since the prior generator is frozen after offline generation. In contrast, the calibrated fused prior $\tilde{P}_i^{\dagger}$ improves from 73.19 to 91.10 IoU, with the best value reaching 91.24 IoU. The selected-region accuracy also increases from 95.10% to 99.68%, while the selected-region coverage remains high and stable from 96.50% to 97.25%. These results verify that L2L progressively refines pseudo supervision through closed-loop calibration and adaptive selection, rather than simply relying on the static MLLM prior.

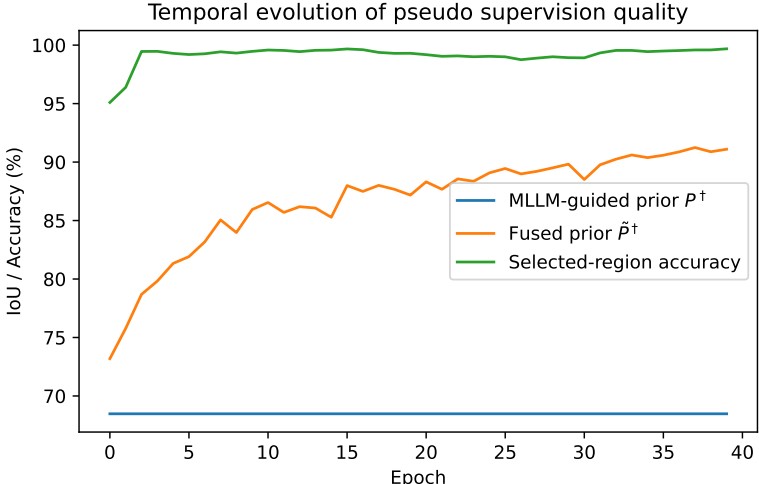

*Figure 9.* Temporal evolution of pseudo supervision quality over the 40-epoch training process. The MLLM-guided prior $P_i^{\dagger}$ remains fixed, while the calibrated fused prior $\tilde{P}_i^{\dagger}$ and the selected-region accuracy progressively improve during training.

**Failure Case Analysis.** As shown in Figure 10, L2L exhibits limitations in complex scenarios. First, **boundary imprecision** persists in challenging environments: severe occlusion causes missing parts, such as the hidden calf's head in Exp-1; intricate structures lead to imprecise foliage delineation in Exp-4; visual clutter obscures fine details like handlebars in Exp-7; and low contrast induces over-segmentation due to high background similarity in Exp-8. Second, **granularity confusion** hinders part-level grounding, where the model shows a "whole-object bias" (e.g., segmenting the entire zebra instead of the "ass" in Exp-2). Third, **high-level reasoning** remains challenging. Despite handling ambiguity (Exp-3), the model falters in spatial and ordinal logic, misinterpreting relative positions (reversing beds in Exp-5) or missing counting constraints (failing to identify the "second" hot dog in Exp-6). Future work will explore stronger reasoning capabilities to better satisfy compositional constraints, including spatial, ordinal, and counting constraints.

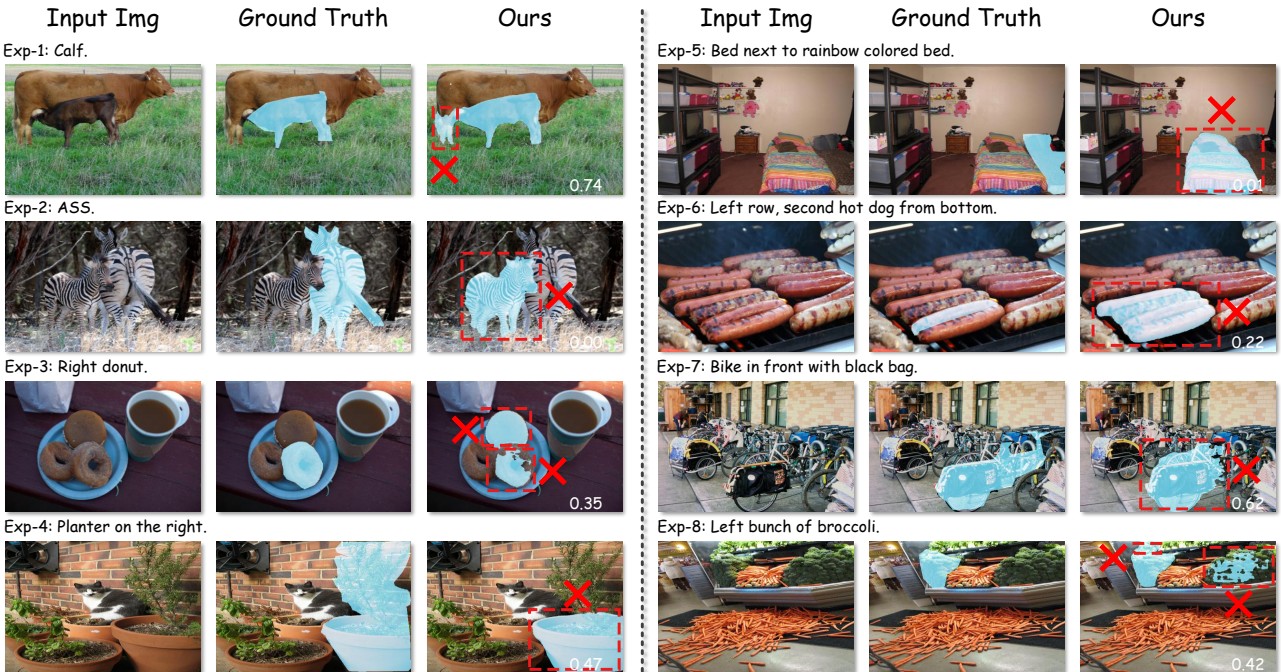

*Figure 10.* **Qualitative analysis of L2L on RefCOCO under the 5% semi-supervised setting.** The white numbers indicate the IoU with the ground-truth mask. Typical failure regions are highlighted with red dashed boxes, and incorrect results are marked in red.

**Additional Qualitative Segmentation Results.**

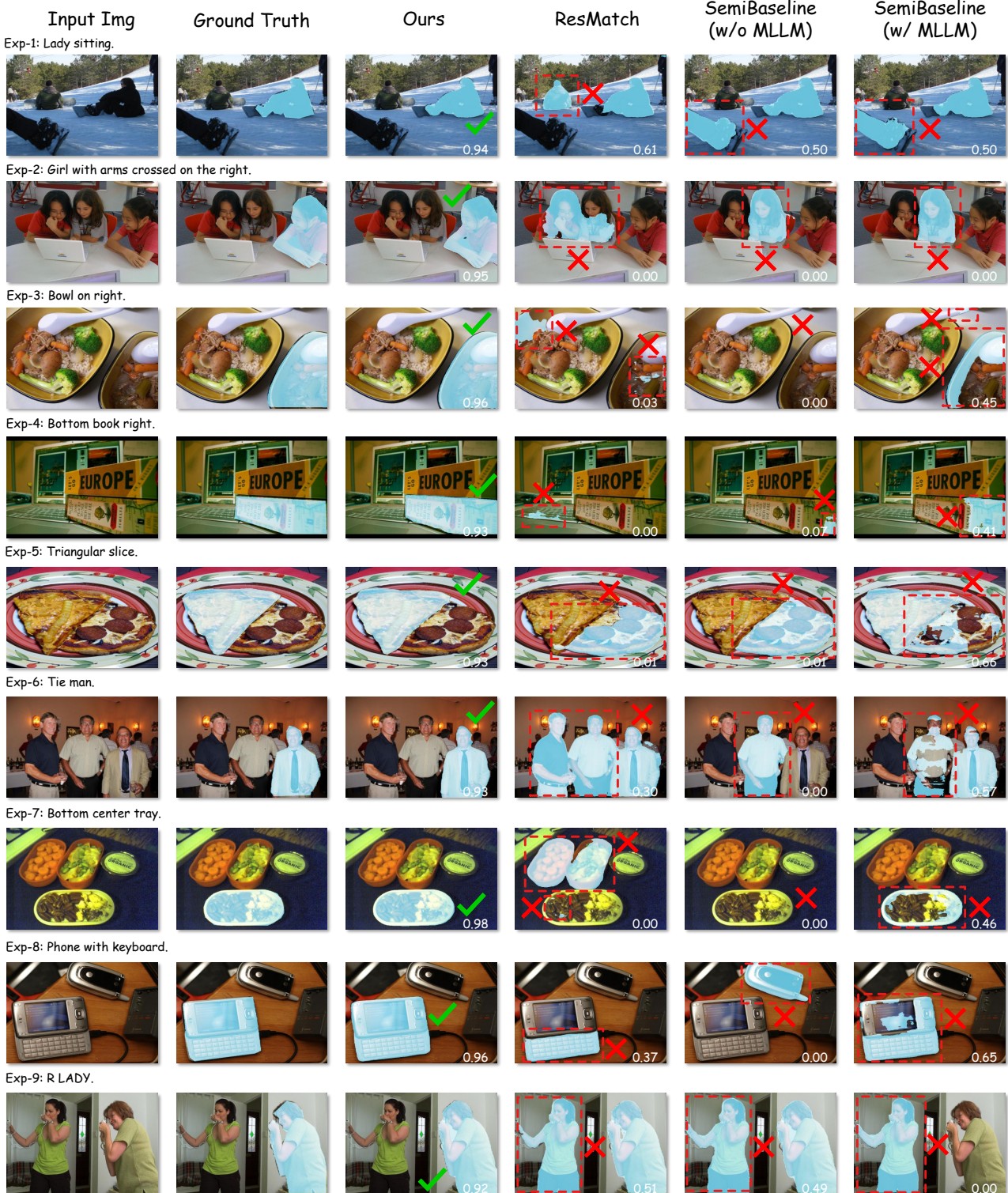

*Figure 11.* **More Qualitative analysis of Ground Truth, L2L, RESMatch, Baseline (w/o MLLM), and Baseline (w/ MLLM) on RefCOCO under the 5% semi-supervised setting.** The white numbers indicate the IoU with the ground-truth mask. Typical failure regions are highlighted with red dashed boxes, and incorrect results are marked in red; correct segmentations are marked in green.

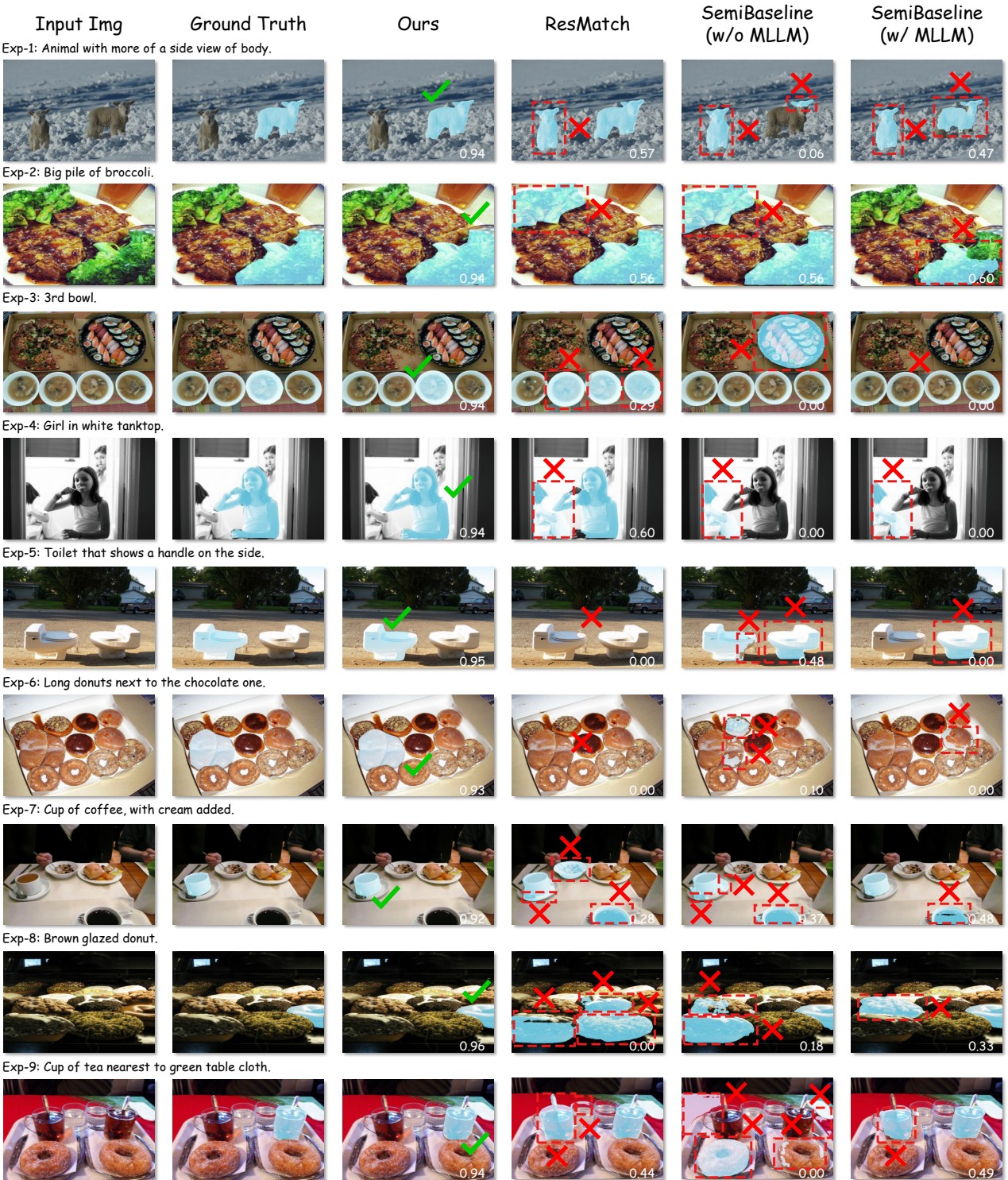

*Figure 12.* **More Qualitative analysis of Ground Truth, L2L, RESMatch, Baseline (w/o MLLM), and Baseline (w/ MLLM) on RefCOCO+ under the 5% semi-supervised setting.** The white numbers indicate the IoU with the ground-truth mask. Typical failure regions are highlighted with red dashed boxes, and incorrect results are marked in red; correct segmentations are marked in green.

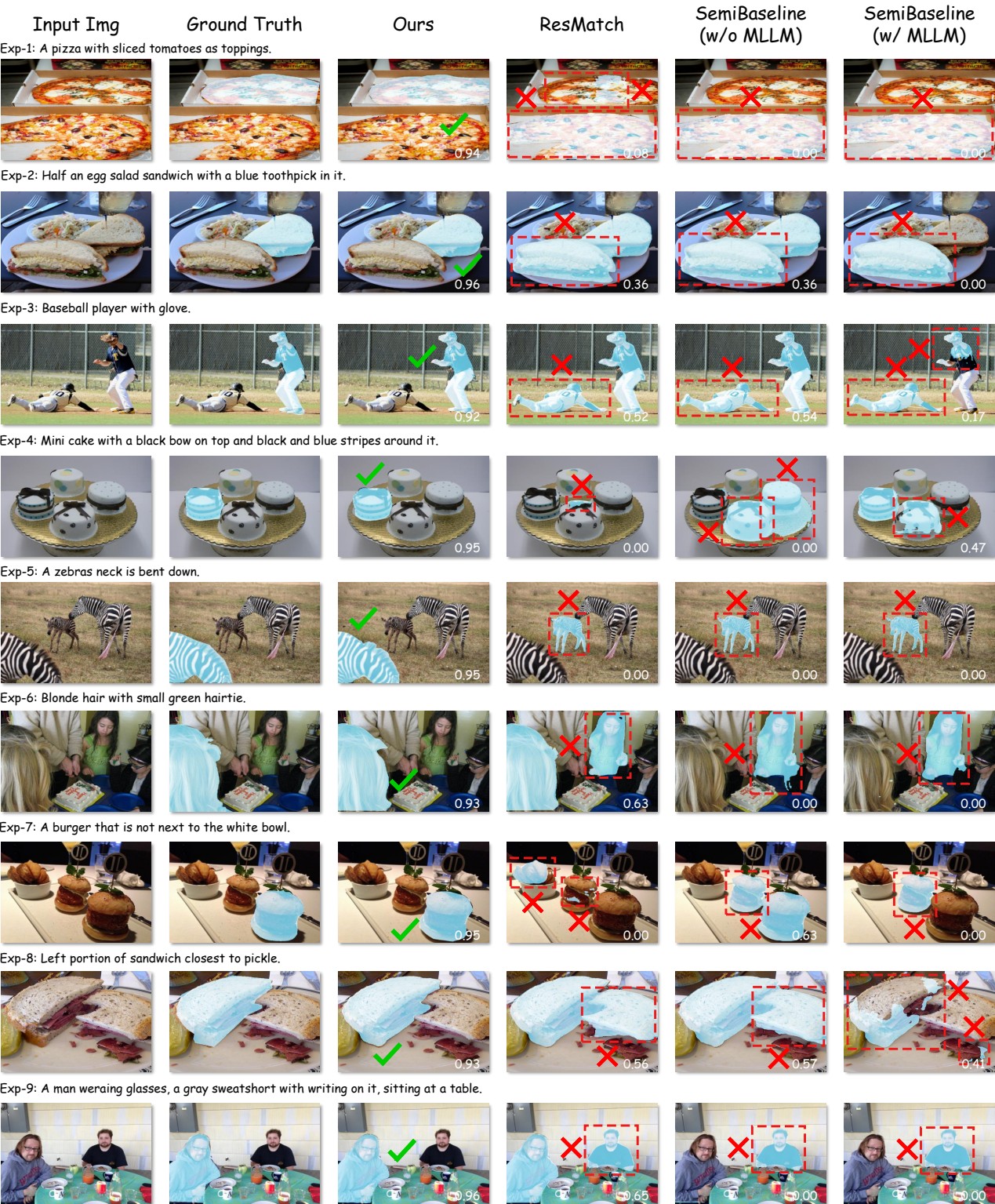

*Figure 13.* **More Qualitative analysis of Ground Truth, L2L, RESMatch, Baseline (w/o MLLM), and Baseline (w/ MLLM) on RefCOCOg under the 5% semi-supervised setting.** The white numbers indicate the IoU with the ground-truth mask. Typical failure regions are highlighted with red dashed boxes, and incorrect results are marked in red; correct segmentations are marked in green.

