# OpenReview forum: "Learning to Label: A Reinforced Self-Evolving Framework for Semi-supervised Referring Expression Segmentation"
_ICML.cc/2026/Conference — ICML 2026 regular_

### Official Review · Reviewer_dNbi · 2026-02-28

**Soundness:** 3
**Presentation:** 3
**Significance:** 3
**Originality:** 2
**Overall Recommendation:** 4
**Confidence:** 4

**Summary:**

This paper introduces Learning to Label (L2L), a reinforced self-evolving framework designed for semi-supervised referring expression segmentation (SS-RES). It innovatively casts pseudo-label construction as a learnable decision-making process, using reinforcement learning (RPLE) to adaptively balance label precision and coverage. To stabilize training under sparse supervision, the framework integrates multimodal priors from MLLMs through an uncertainty-aware calibration module (SPM) and a gated hierarchical segmentation network (SESM). Extensive experiments on RefCOCO, RefCOCO+, and RefCOCOg demonstrate that L2L consistently outperforms existing semi-supervised baselines. Remarkably, with only 0.1% labeled data, the method remains competitive with and often surpasses state-of-the-art zero-shot counterparts

**Compliance With Llm Reviewing Policy:**

Affirmed.

**Final Justification:**

Thank you for the clarification. It addresses some of my concerns, but not sufficiently, so I maintain my original score.

**Key Questions For Authors:**

Please refer to the Weaknesses section.

**Limitations:**

yes

**Strengths And Weaknesses:**

Strengths: The paper successfully replaces rigid, manual heuristics with a Reinforced Pseudo-Label Exploration (RPLE) module that adaptively filters pseudo-labels based on sample difficulty. This technically sound approach effectively addresses the instability of semi-supervised learning in sparse data regimes.The SESM module is well-designed, using a principled gated mechanism to inject MLLM-driven semantic and structural priors into hierarchical encoders. The results are clearly presented, demonstrating that using only 16–21 labeled images can outperform state-of-the-art zero-shot models.

Weaknesses:

1.A primary concern is whether the performance improvements are fundamentally driven by the proposed RL formulation or primarily by the high-quality "pseudo-GT" generated by the frozen MLLM teacher. To properly evaluate the framework's marginal contribution, it is essential to include a naive baseline that directly fine-tunes the segmentor on the raw MLLM outputs, omitting the L2L complexity altogether.


2.The current study relies heavily on highly capable teachers like Qwen2.5-VL. To demonstrate that the RPLE mechanism can genuinely distill useful signals from high-noise environments, the authors should establish a "performance floor." Evaluating the framework with conventional, non-LLM models as the initial teacher (e.g., CARIS [1], CGFormer [2]) would better validate the robustness and generalizability of the pseudo-label filtering.

3.The paper frames the training process as "self-evolving," yet the MLLM remains frozen throughout training, which inherently makes this a one-way knowledge transfer. To substantiate the evolutionary aspect, the authors need to provide a temporal analysis tracking pseudo-label quality over the 40-epoch training span. This would clarify whether the generated pseudo-labels actually refine and improve over time or merely fit to the teacher's static priors.

[1]:Sun-Ao Liu, Yiheng Zhang, Zhaofan Qiu, Hongtao Xie, Yongdong Zhang, and Ting Yao. Caris: Context-aware referring image segmentation. In Proceedings of the 31st ACM International Conference on Multimedia, pp. 779–788, 2023c.
[2]:Jiajin Tang, Ge Zheng, Cheng Shi, and Sibei Yang. Contrastive grouping with transformer for referring image segmentation. In Proceedings of the IEEE/CVF conference on computer vision and pattern recognition, pp. 23570–23580, 2023.

---

> ### Author Rebuttal · Authors · 2026-03-30
>
> We thank the reviewer for the comments.
>
>  **Q1: The authors should include a naive baseline that directly fine-tunes the segmenter on the original MLLM outputs, without introducing the L2L design.**
>
> **A1:** Thanks for your valuable comment. To directly assess whether the gain mainly comes from the frozen MLLM teacher or from the proposed framework itself,  we add a direct teacher-only baseline by removing L2L design and training the segmentor on raw MLLM pseudo labels, **as shown in Table I**. Since the teacher source is unchanged, this comparison isolates the effect of pseudo-label construction. L2L method performs better across all splits, indicating that the gain is not explained by the stronger teacher alone.
>
> **Table I. Comparison with a teacher-only baseline on RefCOCO under the 5% labeled setting.**
> |Method|val|testA|testB|
> |---|---:|---:|---:|
> |Raw MLLM pseudo labels|65.69|70.26|60.72|
> |Ours|**67.30**|**71.46**|**62.60**|
>
> **Table II further compares stronger heuristic baselines under the same prior source.** Even with tuned mixing, agreement-based filtering, or adaptive thresholding [3], our full method still achieves better performance. Taken together, Tables I and II show that the gain is not explained by the frozen teacher alone or by simple prior-exploitation heuristics, but by the learnable calibration and selection mechanism in L2L.
>
> **Table II. Performance comparison with other heuristic methods (RefCOCO, 5\% labeled).**
> |Method|val|testA|testB|
> |---|---:|---:|---:|
> |Baseline + Agreement filtering |64.30|68.98|58.59|
> |Baseline + CATM|65.05|69.50|59.90|
> |Ours|**67.30**|**71.46**|**62.60**|
>
> **Q2: ... the L2L complexity.**
>
> **A2:** The main additional cost of our framework comes from one-time offline prior generation, not repeated end-to-end optimization over foundation models. The MLLM-guided and SAM priors are generated once for the unlabeled set, remain frozen during training; inference uses only the segmentation network $S_\theta$. The RL branch is lightweight with a 6-dimensional state and a 2-dimensional action space, and the actor/critic together contain only 0.009539M trainable parameters. It is updated once per unlabeled mini-batch during training and removed entirely at inference time. **As shown in Table III**, compared with the MLLM-based baseline, our method introduces only 9.353M additional trainable parameters.
>
> **Table III. Comparison of trainable and total parameters.**
> |Method|Trainable Params (M)|Total Params (M)|
> |---|---:|---:|
> |Baseline (w/o MLLM)|227.740|227.740|
> |Baseline (w/ MLLM)|227.740|8744.337|
> |Ours|237.093|8753.690|
>
> **Q3: Non-LLM models as the initial teacher would better validate the robustness and generalizability of the pseudo-label filtering.**
>
> **A3:** We thank the reviewer for this valuable suggestion. To directly evaluate it, we replace the original MLLM-based prior with priors generated by conventional non-LLM teachers (CARIS [1] and CGFormer [2]), while keeping the rest of the L2L unchanged. **As shown in Table IV**, the resulting performance remains strong and broadly comparable to the original version, suggesting that L2L is not restricted to MLLM-generated priors. This is also consistent with our formulation: L2L operates on a generic external prior $P_i^\dagger$, which is calibrated by SPM and then exploited by SESM and RPLE. Hence, the framework is not tied to a specific MLLM teacher, but provides a general mechanism for turning different prior sources into reliable supervision.
>
> **Table IV. L2L with conventional non-LLM teachers on RefCOCO under the 5% labeled setting.**
> |Method|val|testA|testB|
> |---|---:|---:|---:|
> |with [1] prior|67.04|70.80|**64.02**|
> |with [2] prior|66.46|70.42|62.82|
> |Ours|**67.30**|**71.46**|62.60|
>
> **Q4: Authors need to provide a temporal analysis tracking pseudo-label quality over the 40-epoch training span.**
>
> **A4:** We further substantiate the “self-evolving” aspect with a temporal analysis over 40 epochs (**see https://anonymous.4open.science/r/L2L-34F7/image.png**). While the MLLM-guided prior $P^\dagger$ remains fixed at 68.48 IoU because the MLLM is frozen, the pseudo supervision used by L2L improves steadily: fused prior $\tilde{P}^\dagger$ increases from 73.19 to 91.10 IoU (best 91.24), selected-region accuracy rises from 95.10% to 99.68%, and coverage stays high and stable (96.50% $\rightarrow$ 97.25%). The learned foreground threshold $\tau_t^{fg}$ also evolves during training, indicating that the policy adapts rather than merely reproduces the static teacher prior. Therefore, in our framework, “self-evolving” refers to progressively refined pseudo supervision through closed-loop calibration and adaptive selection, rather than to updating the frozen MLLM itself.
>
> [1] Caris:Context-aware referring image segmentation, MM 2023.
>
> [2] Contrastive grouping with transformer for referring image segmentation, CVPR 2023.
>
> [3] Adaptive self-training framework for fine-grained scene graph generation, ICLR 2024.

---

> > ### Author Rebuttal · Reviewer_dNbi · 2026-04-02
> >
> > While the rebuttal helpfully clarifies several points and partially alleviates my concerns regarding teacher dependence and the “self-evolving” claim, it does not yet fully isolate the contribution of the RL formulation itself or convincingly establish robustness under genuinely weak or noisy teachers, so I maintain my original score.

---

> > > ### Author Response · Authors · 2026-04-07
> > >
> > > Thank you for your prompt response. We are pleased to hear that our previous reply addressed your concerns. We will now address your new questions one by one.
> > >
> > > **Q1: It does not yet fully isolate the contribution of the RL formulation itself.**
> > >
> > > **A1:** Thanks once again for your valuable feedback. To more directly assess the specific contribution of RPLE within the full framework, we replace only RPLE while keeping the prior source, SPM, SESM, backbone, and training protocol unchanged. **We compare against two heuristic dynamic thresholding methods: Curriculum and CATM [1]. As shown in Table I**, the full model with RPLE consistently achieves the best performance. This comparison isolates the effect of the decision mechanism itself, and shows that the improvement is not explained by adaptive thresholding alone, but by the RL-based policy search in RPLE.
> > >
> > > **Table I. Replacing RPLE with dynamic thresholding methods on RefCOCO (5\% labeled).**
> > > |Method|val|testA|testB|
> > > |---|---:|---:|---:|
> > > |Replace RPLE with Curriculum|65.69|69.43|60.48|
> > > |Replace RPLE with CATM|65.86|70.90|60.77|
> > > |Ours|**67.30**|**71.46**|**62.60**|
> > >
> > > We further add a reward ablation to clarify that the gain is not from extra parameters or arbitrary threshold perturbation, but from the specific RL objective. In RPLE, the reward is formulated as a multi-objective signal consisting of **separability gain** ($\Delta M$), **consistency gain** ($\Delta K$), and **coverage gain** ($\Delta C$). We examine the contribution of these three reward components through a component-wise study. **As shown in Table II,** each component contributes to the final performance, and the full reward achieves the best results. This indicates that the effectiveness of RPLE comes from jointly optimizing pseudo-label quality along these complementary dimensions rather than from a generic dynamic-threshold effect.
> > >
> > > **Table II. Reward-component ablation for RPLE on RefCOCO (5\% labeled)**
> > > |Method|val|testA|testB|
> > > |---|---:|---:|---:|
> > > |w/o $\Delta M$|66.35|70.21|62.54|
> > > |w/o $\Delta K$|65.36|69.12|60.88|
> > > |w/o $\Delta C$|65.95|69.64|61.56|
> > > |Ours|**67.30**|**71.46**|**62.60**|
> > >
> > > Overall, these results more directly isolate the contribution of the RL formulation: RPLE is a learned policy that optimizes pseudo-label quality through separability, consistency, and coverage jointly.
> > >
> > > **Q2: Establish robustness under genuinely weak or noisy teachers.**
> > >
> > > **A2**: Thanks again for your careful comments. To more directly address this concern, we further examine our framework under both **weak** and **noisy** teachers.
> > >
> > > **For weak teachers**, we evaluate robustness from two complementary perspectives. **First, we vary the segmentor while keeping the MLLM fixed. As shown in Table III**, using SAM instead of SAM2 still yields comparable performance, suggesting that the framework is not tightly coupled to a specific segmentation prior generator. **Second, we vary the MLLM teacher while keeping the segmentor fixed.** Using the weaker Qwen2.5-VL-3B in place of Qwen2.5-VL-7B also gives broadly comparable results, indicating that the framework does not rely on a single strong MLLM teacher. This is consistent with our design: L2L further restricts pseudo supervision to selected trustworthy regions through sample-adaptive filtering. Therefore, the framework is designed not to depend on a particularly strong teacher, but to handle priors through calibration and adaptive selection.
> > >
> > > **Table III. Foundation-model sensitivity under the RefCOCO 5\% labeled setting.**
> > > |MLLM backbone|Segmentor|val|testA|testB|
> > > |---|---|---:|---:|---:|
> > > |Qwen2.5-VL-7B|SAM2|**67.30**|**71.46**|**62.60**|
> > > |Qwen2.5-VL-7B|SAM|66.77|71.23|61.23|
> > > |Qwen2.5-VL-7B|SAM2|**67.30**|**71.46**|62.60|
> > > |Qwen2.5-VL-3B|SAM2|66.64|71.09|**62.96**|
> > >
> > > **For noisy teachers**, we further conduct an explicit prior-corruption experiment. Starting from the Qwen-generated box, we inject synthetic noise before SAM prompting: with 50\% probability we apply a random translation, and with 50\% probability we apply a random scaling, where the perturbation magnitude is uniformly sampled from 1\% to 30\% of the box size. The box is fed into SAM to generate a degraded prior. **As shown in Table IV**, introducing noise into the teacher prior leads to a moderate performance drop compared with the clean setting, but the noisy-prior variant still remains clearly stronger than the baseline. This suggests that the proposed calibration-and-selection mechanism remains effective even when the teacher signal is explicitly degraded.
> > >
> > > **Table IV. Robustness under noisy teachers on RefCOCO (5\% labeled).**
> > > |Method|val|testA|testB|
> > > |---|---:|---:|---:|
> > > |Baseline (w/ MLLM)|64.07|69.65|58.37|
> > > |Noisy prior|66.13|70.18|61.13|
> > > |Ours|**67.30**|**71.46**|**62.60**|
> > >
> > > Thank you again for your valuable comments and thoughtful questions. We hope that our responses can address your concerns.
> > >
> > > [1] Adaptive self-training framework for fine-grained scene graph generation, ICLR 2024.

---

### Official Review · Reviewer_3vrB · 2026-03-09

**Soundness:** 3
**Presentation:** 3
**Significance:** 3
**Originality:** 2
**Overall Recommendation:** 4
**Confidence:** 3

**Summary:**

This paper proposes L2L, a reinforced self-evolving framework for semi-supervised referring expression segmentation. To address the pseudo-label noise caused by traditional fixed thresholds and MLLM prior mismatches, this method generates initial spatial priors using MLLMs and SAM2 with uncertainty calibration. It then introduces a reinforcement learning mechanism to dynamically output adaptive pseudo-label filtering thresholds based on sample states. Under extremely low annotation budgets (0.1% to 10%), L2L outperforms existing baselines in both segmentation accuracy and robustness.

**Compliance With Llm Reviewing Policy:**

Affirmed.

**Final Justification:**

The authors' rebuttal has solved my concerns, and I keep my original positive rating based on the paper's strengths.

**Key Questions For Authors:**

See weaknesses.

**Limitations:**

yes

**Strengths And Weaknesses:**

**Strengths**

The proposed self-evolving framework enables the joint iterative evolution of the threshold selection strategy and the segmentation network with minimal parameter overhead, achieving significant performance gains on representative RES datasets.

By utilizing a staged, dynamically weighted SESM module and modeling pseudo-label filtering as an RL-driven decision process, the method effectively incorporates multimodal priors and consistency information. This addresses the common pitfalls of static thresholds, such as sample wastage from being too conservative or noise accumulation from being too aggressive.

**Weaknesses**

The evaluation of RPLE is limited to comparisons against static thresholds. Since semi-supervised learning includes lightweight heuristic dynamic methods (e.g., curriculum-based threshold increments), outperforming a naive fixed baseline is insufficient to justify the necessity of a complex RL mechanism.

L2L leverages Qwen2.5-VL 7B and SAM2, whereas prior SOTA methods like SemiRES or RESMatch rely on SAM1 or no foundation models at all. This raises concerns about whether the reported performance gains come from the proposed framework or simply the use of more powerful foundation models.

The reward function in the RPLE module contains several key hyperparameters that lack systematic ablation or sensitivity analysis, which weakens the arguments regarding robustness and reproducibility. Furthermore, the joint optimization of the discriminator loss $L_{critic}$ and the primary segmentation loss is not clearly explained.

The "stage-alignment operator" in the SESM module appears to be a custom heuristic rather than a standard module. The text explains its role in aligning priors to feature resolutions across stages but provides insufficient detail regarding its specific implementation.

---

> ### Author Rebuttal · Authors · 2026-03-30
>
> We thank the reviewer for the positive assessment and the constructive feedback.
>
> **Q1: The evaluation of RPLE is limited to comparisons against static thresholds.**
>
> **A1:** Following your suggestion, we replace only RPLE while keeping all other components fixed for a fair comparison, and compare with two dynamic baselines: a curriculum schedule and CATM [1]. **As shown in Table I,** our RPLE-based model consistently achieves better performance, demonstrating its effectiveness for pseudo-label construction beyond standard adaptive thresholding.
>
> **Table I. Comparison with dynamic baselines by replacing RPLE on RefCOCO (5% labeled).**
> |Method|val|testA|testB|
> |---|---:|---:|---:|
> |Replace RPLE with Curriculum|65.69|69.43|60.48|
> |Replace RPLE with CATM|65.86|70.90|60.77|
> |Ours|**67.30**|**71.46**|**62.60**|
>
> **Q2: Performance gains come from the proposed framework or simply the use of more powerful foundation models.**
>
> **A2:** Thanks for reviewing. While stronger foundation priors do contribute, the observed gains cannot be attributed to the prior alone. First, our sensitivity study (Appendix Table 10) shows that replacing the MLLM backbone (7B → 3B) or the segmentor leads to only minor performance changes, indicating that our framework does not rely on a specific strong prior. Second, **as shown in Table II,** even with stronger baselines (e.g., tuned mixing, agreement-based filtering, and adaptive thresholding), our method still achieves superior performance. These results suggest that the gains mainly come from our framework, which more effectively calibrates and leverages imperfect priors, rather than from the prior source itself.
>
> **Table II. Stronger prior-exploitation baselines under the same prior source.**
> |Method|val|testA|testB|
> |---|---:|---:|---:|
> |Baseline + Tuned mixing|64.59|68.97|59.71|
> |Baseline + Agreement filtering|64.30|68.98|58.59|
> |Baseline + Adaptive-threshold baseline|65.05|69.50|59.90|
> |Ours|**67.30**|**71.46**|**62.60**|
>
> **Q3: The reward function in the RPLE module contains several key hyperparameters.**
>
> **A3:** To strengthen the robustness analysis of the reward design and clarify the optimization procedure, **we provide additional ablations in Table III.** Specifically, we conduct component-wise studies (w/o $\Delta M$, w/o $\Delta K$, and w/o $\Delta C$) to assess whether RPLE’s gains rely on any single reward term. As the exploratory reward is formulated as a multi-objective signal capturing separability, consistency, and coverage, removing each component in turn offers a direct evaluation of its contribution.
>
> **Table III. Reward-component ablation for RPLE on RefCOCO (5% labeled).**
> |Method|val|testA|testB|
> |---|---:|---:|---:|
> |w/o $\Delta M$|66.35|70.21|62.54|
> |w/o $\Delta K$|65.36|69.12|60.88|
> |w/o $\Delta C$|65.95|69.64|61.56|
> |Ours|**67.30**|**71.46**|**62.60**|
>
> **Q4: The joint optimization of the discriminator loss ${L}_{critic}$ and the primary segmentation loss is not clearly explained.**
>
> **A4:** Within each training iteration, the RL branch ($L_{critic}$) and the segmentation branch ($L$, Appendix B.3) are updated sequentially. Specifically, the Critic $Q_{\varphi}(s,a)$ and the policy searcher $\pi_{\phi}(s_t)$ are first updated from the transition $(s_t,a_t,r_t,s_{t+1})$ using the critic objective, so as to learn an adaptive pseudo-label selection policy. The segmentation network is then optimized on the RPLE-selected regions using the overall objective $L$. Therefore, the two branches are not jointly optimized through a single mixed loss, but interact through pseudo-label selection.
>
> **Q5: The "stage-alignment operator" in the SESM module appears to be a custom heuristic and provides insufficient detail regarding its specific implementation.**
>
> **A5:** We thank the reviewer for the comment. Concretely, the prior mask is first resized by bilinear interpolation to the spatial resolution of each encoder stage, and then projected to the corresponding stage channel dimension by a $1\times1$ convolution to form the aligned structural field. The subsequent geometric interaction is stage-dependent: shallow stages use projection-based gated injection to preserve local spatial correspondence, while deeper stages use attention-based interaction over the aligned structural field to enable relational reasoning. The operator is therefore a concrete implementation of the stage-dependent $\Psi_{geo}^{(j)}$ in Eq. (2). We will revise the paper to make these implementation steps and their connection to Eqs. (2)--(3) more explicit.
>
> [1] Adaptive self-training framework for fine-grained scene graph generation, ICLR 2024.

---

> > ### Author Rebuttal · Reviewer_3vrB · 2026-04-03
> >
> > Thanks for the rebuttal and I will keep my positive rating.

---

> > > ### Author Response · Authors · 2026-04-06
> > >
> > > Dear Reviewer 3vrB,
> > >
> > > We sincerely thank you for your positive response and for taking the time to review our article. We are truly grateful that our clarifications and additional efforts have adequately addressed your concerns.
> > >
> > > We truly appreciate your constructive and insightful feedback. Your comments have provided us with important guidance for improving the manuscript and have played a valuable role in strengthening the quality of our work. We will carefully consider each point you raised and address them comprehensively in the revised version.
> > >
> > > Thank you once again for your time and valuable feedback.
> > >
> > > Sincerely,
> > >
> > > Authors

---

### Official Review · Reviewer_MVDu · 2026-03-10

**Soundness:** 3
**Presentation:** 3
**Significance:** 3
**Originality:** 3
**Overall Recommendation:** 5
**Confidence:** 4

**Summary:**

This paper addresses the issues of scarce labeling and unreliable pseudo-labels in the semi-supervised referring expression segmentation (SS-RES) task, and proposes the Learning to Label (L2L) framework. This method first utilizes a multimodal large language model (MLLM) to extract semantic-spatial priors to initialize soft segmentation proposals. Subsequently, it innovatively introduces the Reinforce Pseudo-label Exploration (RPLE) mechanism, modeling the selection of pseudo-labels as an adaptive decision-making process, to mine high-value pixel-level supervision signals. Experimental results on the RefCOCO, RefCOCO+, and RefCOCOg datasets demonstrate that this method significantly outperforms existing semi-supervised segmentation methods.

**Compliance With Llm Reviewing Policy:**

Affirmed.

**Key Questions For Authors:**

See weaknesses.

**Limitations:**

Yes

**Strengths And Weaknesses:**

**Strengths:**
1. The article is well-structured, with clear and understandable content, a clear motivation, and a complete writing framework.
2. The work formalizes the construction and selection process of pseudo-labels as a learnable reinforcement learning decision problem, rather than the traditional static threshold filtering. This is an insightful innovation that can adaptively adjust the supervision signals based on the difficulty of the samples.
3. Extensive experiments were conducted on three mainstream benchmark datasets, and significant performance improvements were achieved, demonstrating the effectiveness and generalization ability of the framework under different supervision ratios.

**Weaknesses:**
1. As shown in Figure 3, during the training process, the MLLM-Guided Semantic Prior needs to be frequently called, and a complex RL strategy searcher needs to be run. Considering the inference delay of MLLM, this training paradigm has a huge computational cost (more than 8B parameters).
2. The core of the RPLE module lies in the design of exploratory reward. In the early training stage, the model's predictions may be very inaccurate, and how does it avoid being misled by noise and choose incorrect pseudo-labels at this time?
3. This method highly relies on the initial priors generated by MLLM. If MLLM generates hallucinations or provides an incorrect bounding box in complex scenarios, will the subsequent SPM calibration and RL filtering have sufficient error-correction capabilities?

---

> ### Author Rebuttal · Authors · 2026-03-30
>
> We thank the reviewer for the positive assessment and the insightful concerns on practicality and robustness.
>
> **Q1: Concern about the training cost introduced by the MLLM-guided prior and RL-based strategy search.**
>
> **A1:** Thank you for reviewing. In our work, **the MLLM-guided and  SAM priors are generated offline once for the unlabeled set and are not repeatedly queried during end-to-end training**. In other words, the main additional cost is a one-time preprocessing step, rather than per-iteration or per-epoch optimization over foundation models. Importantly, inference uses only the segmentation network $S_\theta$, without requiring either MLLM or SAM. Regarding the RL component, the additional overhead of RPLE itself is lightweight. In our implementation, both the actor and critic are simple two-layer MLPs with hidden size 64, so the RL module adds only a modest training cost compared with the segmentation backbone. **We report the average offline prior-generation time per image--text pair, measured on NVIDIA RTX A6000 GPU in Table I, and we further report the trainable/total parameter comparison to clarify that the extra cost does not come from end-to-end training of a very large foundation model in Table II.** Compared with the MLLM-based baseline, our method introduces only 9.353M additional trainable parameters. Overall, these results indicate that the extra cost of our framework mainly comes from one-time offline preprocessing, rather than end-to-end training of a very large model.
>
> **Table I. Average offline prior-generation time per image--text pair.**
> |Metric|Qwen2.5-VL-7B|SAM1|SAM2|
> |---|---:|---:|---:|
> |Avg. time (s/pair)|1.7430|0.4541|0.1661|
>
> **Table II. Comparison of trainable and total parameters.**
> |Method|Trainable Params (M)|Total Params (M)|
> |---|---:|---:|
> |Baseline (w/o MLLM)|227.740|227.740|
> |Baseline (w/ MLLM)|227.740|8744.337|
> |Ours|237.093|8753.690|
>
> **Q2: In the early training stage, the model's predictions may be very inaccurate, and how does it avoid being misled by noise and choose incorrect pseudo-labels at this time?**
>
> **A2:** We thank the reviewer for this important concern. We agree that early-stage pseudo-labeling is particularly vulnerable to noise, and our design therefore makes RPLE deliberately conservative at the beginning of training, rather than relying on unconstrained exploration. Specifically, several mechanisms are introduced to reduce the risk of noisy predictions corrupting the reward signal.
> 1. **SPM-based prior calibration.**  SPM calibrates the external prior with the weak-view prediction and down-weights the prior when it conflicts with the model or is deemed unreliable.
> 2. **Conservative pseudo-label selection.** RPLE is initialized with conservative thresholds $(\tau_{fg}=0.70,\tau_{bg}=0.20)$ and ignores a 3-pixel boundary band, so that highly ambiguous regions are excluded from supervision in the early stage.
> 3. **Stabilized exploratory reward.** The exploratory reward is regularized by incorporating a stability penalty and applying clipping to $[-0.5, 0.5]$, which helps stabilize policy learning, especially when the segmentation model is still immature. In addition, RPLE is lightweight, with both the actor and critic implemented as two-layer MLPs (hidden size 64), which further limits the risk of unstable policy updates. Overall, RPLE does not depend purely on raw early predictions, but instead leverages calibrated priors together with a conservative reward design to ensure stable learning.
>
> **Q3: If MLLM generates hallucinations or provides an incorrect bounding box in complex scenarios, will the subsequent SPM calibration and RL filtering have sufficient error-correction capabilities?**
>
> **A3:** We agree that raw MLLM priors can be imperfect in challenging scenes. Our framework is therefore designed not to directly trust the prior, but to calibrate and filter it before it is used as pseudo supervision. SPM explicitly down-weights the prior when it conflicts with the weak-view prediction or appears unreliable, and RPLE further restricts supervision to selected trustworthy regions through sample-adaptive filtering. Thus, the role of SPM and RPLE is not to assume perfect priors, but to reduce the chance that erroneous priors are propagated into training. We do not claim that these components can fully correct arbitrary hallucinations or severely wrong grounding in every complex case. Rather, they are intended to make the framework less sensitive to imperfect priors. **This is consistent with our foundation-model sensitivity result in Table III**, where replacing SAM2 with SAM or Qwen2.5-VL-7B with Qwen2.5-VL-3B causes only minor performance changes.
>
> **Table III. Foundation-model sensitivity under the RefCOCO 5% labeled setting.**
> |MLLM backbone|Segmentor|val|testA|testB|
> |---|---|---:|---:|---:|
> |Qwen2.5-VL-7B|SAM2|67.30|71.46|62.60|
> |Qwen2.5-VL-7B|SAM|66.77|71.23|61.23|
> |Qwen2.5-VL-3B|SAM2|66.64|71.09|62.96|

---

> > ### Author Rebuttal · Reviewer_MVDu · 2026-04-05
> >
> > I keep my score. Looking forward to the code being made open source.

---

> > > ### Author Response · Authors · 2026-04-06
> > >
> > > Dear Reviewer MVDu,
> > >
> > > We sincerely thank you for your encouraging acknowledgement and for taking the time to carefully review our rebuttal. We are very grateful to know that our response has adequately addressed your concerns.
> > >
> > > We also greatly appreciate your positive assessment of our work. In particular, we thank you for your interest in the open-source release of our code. We plan to make the code open source to support transparency, reproducibility, and future research.
> > >
> > > Thank you once again for your time, support, and valuable feedback.
> > >
> > > Sincerely,
> > >
> > > Authors

---

### Official Review · Reviewer_Vp1K · 2026-03-11

**Soundness:** 4
**Presentation:** 3
**Significance:** 2
**Originality:** 3
**Overall Recommendation:** 4
**Confidence:** 4

**Summary:**

This paper proposes Learning to Label (L2L), a framework for semi-supervised referring expression segmentation (SS-RES). The method uses a frozen MLLM (Qwen2.5-VL) to generate bounding box predictions for unlabeled image–text pairs, which are then fed into a frozen SAM2 to produce soft segmentation priors. These external priors are fused with the segmentation model's own predictions via an uncertainty-aware module (SPM), injected into the encoder via stage-adaptive gating (SESM), and used to generate pseudo-labels whose pixel-level selection is governed by a DDPG-based reinforcement learning agent (RPLE). The actual segmentation network (Swin Transformer + BERT) is the only component trained end-to-end. Experiments on RefCOCO, RefCOCO+, and RefCOCOg under 0.1%–10% label budgets show improvements over existing SS-RES methods.

**Compliance With Llm Reviewing Policy:**

Affirmed.

**Final Justification:**

The major concern of this paper was 1) the most of performance gain comes the architecture selection (MLLM + SAM), and 2) lack of justification of using offline RL, which requires a lot of computational cost compared to the existing pseudo-labels. The authors have resolved this issue by providing new experiments, addressing my concern accordingly.

**Key Questions For Authors:**

- Can you add natural heuristic baselines—using pseudo-labels only where both the MLLM prior and model prediction are confident (intersection-based filtering).

- Given that the state space is 6D and action space is 2D, have you compared RPLE against simpler adaptive thresholding baselines?

- The Baseline (w/ MLLM) uses a fixed mixing coefficient of 0.5. What happens when this coefficient is optimized via grid search on the validation set?

**Limitations:**

yes

**Strengths And Weaknesses:**

## Strengths

- **Well-motivated observation.** The analysis of confidence mismatch between MLLM priors and model predictions (Figure 1) is insightful and clearly illustrates why fixed-threshold pseudo-labeling is suboptimal for this task. The scatter plot effectively demonstrates the sample-dependent nature of disagreement.

- **Practically relevant problem.** Reducing annotation cost for RES is genuinely important, and the semi-supervised formulation is a natural direction. The paper clearly articulates why standard SSL methods struggle with referring expression segmentation due to referential ambiguity and boundary uncertainty.

- **Clean framework design.** The overall pipeline—external prior generation, uncertainty-aware fusion, conditional guidance injection, and adaptive pseudo-label selection—is logically coherent and well-structured.


## Weakness

- The dominant source of improvement is the external foundation model, not the proposed method. This is the most critical concern. Table 1 reveals that the overwhelming majority of performance gain comes from simply using MLLM+SAM2 priors, not from the proposed modules (SPM, SESM, RPLE). For further rigorous demonstration, more advanced heuristic should be added to the baseline. In my opinion, the common pseudo-label method without MLLM little provide the information to demonstrate the effectiveness of the proposed methods Specifically, the Baseline (w/ MLLM) uses a fixed mixing coefficient of 0.5, which is an intentionally naive design. I suggest to include (a) Baseline (w/ MLLM) with a tuned mixing coefficient, (b) agreement-based filtering (use pseudo-labels only where both MLLM and model are confident), (c) any adaptive thresholding method (e.g., FlexMatch, CATM from ST-SGG [1, 2]). These comparison will provide how good the proposed modules are when doing semi-supervised learning.


- I have concern on the RL formulation (RPLE) in terms of the problem's complexity. The state space is 6-dimensional and the action space is 2-dimensional. DDPG is designed for high-dimensional continuous control problems—using it for a 6 -> 2 mapping is excessive. The authors should claim what is the advantage of using RL approaches compared to the simple adaptive thresholding like [1, 2].

- While MLLM and SAM2 are used offline, the overhead of generating priors for all unlabeled data, plus the RL agent's training cost, is never reported.

[1] FlexMatch: Boosting Semi-Supervised Learning with Curriculum Pseudo Labeling

[2] Adaptive Self-training Framework for Fine-grained Scene Graph Generation

---

> ### Author Rebuttal · Authors · 2026-03-30
>
> We thank the reviewer for the constructive feedback and for recognizing the motivation, practical relevance and clear design of L2L.
>
> **Q1: More advanced heuristic should be added to the baseline.**
>
> **A1:** Thanks for this valuable comment. To more rigorously assess whether the gain comes from the external prior alone or from how it is exploited, we further add three stronger MLLM-based baselines for comparison: **1) the baseline (w/ MLLM) with tuned mixing coefficients; 2) the agreement-based filtering (threshold > 0.7); 3) an adaptive thresholding method (such as CATM [2]).**
>
> **As shown in Table I**, while careful tuning of the mixing coefficient $\alpha, \beta$ yields moderate improvements on several splits, we adopt a fixed value of 0.5 as a simple and consistent baseline for all experiments. More importantly, **Table II shows that even with stronger strategies such as agreement-based filtering and CATM**, L2L still consistently outperforms them. This suggests that improved heuristics alone are insufficient to fully exploit the prior, whereas our method provides a more effective way to handle sample-dependent prior–prediction mismatch. We will revise the paper to include these stronger baselines and make the attribution analysis more explicit.
>
> **Table I. Effect of tuned mixing coefficients on Baseline (w/ MLLM) (RefCOCO, 5\% labeled).**
> |$\alpha$|$\beta=1-\alpha$|val|testA|testB|
> |:---:|:---:|---:|---:|---:|
> |0.1|0.9|60.96|67.07|55.15|
> |0.3|0.7|64.55|68.97|**59.71**|
> |0.5|0.5|64.07|**69.65**|58.37|
> |0.7|0.3|64.49|68.27|58.72|
> |0.9|0.1|**64.59**|67.95|58.97|
>
> **Table II. Performance comparison with other heuristic methods (RefCOCO, 5% labeled).**
> |Method|val|testA|testB|
> |---|---:|---:|---:|
> |Baseline + Agreement filtering |64.30|68.98|58.59|
> |Baseline + CATM|65.05|69.50|59.90|
> |Ours|**67.30**|**71.46**|**62.60**|
>
> **Q2: The authors should claim what is the advantage of using RL approaches compared to the simple adaptive thresholding like [1, 2].**
>
> **A2:** Thanks for the suggestion. To directly compare RL with adaptive thresholding, we replace only the RPLE module with two representative adaptive methods (CATM [1] and curriculum thresholding), while keeping all other components unchanged for a fair comparison. **As shown in Table III**, our full model with RPLE consistently achieves the best performance. This indicates that simply adopting adaptive thresholding is insufficient, whereas RPLE provides a more effective solution for pseudo-label selection. We will revise the paper to better highlight this advantage.
>
> **Table III. Comparison between RPLE and simpler adaptive-threshold alternatives on RefCOCO.**
> |Method|val|testA|testB|
> |---|---:|---:|---:|
> |Replace RPLE with CATM|65.86|70.90|60.77|
> |Replace RPLE with Curriculum|65.69|69.43|60.48|
> |Ours|**67.30**|**71.46**|**62.60**|
>
> **Q3: While MLLM and SAM2 are used offline, the overhead of generating priors for all unlabeled data, plus the RL agent's training cost, is never reported.**
>
> **A3**: Thanks for your valuable comment. The main additional cost of our framework comes from one-time offline prior generation, rather than per-epoch end-to-end optimization over foundation models. Specifically, the MLLM and SAM priors are generated once for the unlabeled set, remain frozen, and are not involved in backpropagation during training. **The associated offline prior-generation cost, measured on NVIDIA RTX A6000 GPU, is summarized in Table IV.**
>
> The RL branch is implemented as a lightweight DDPG module with a 6-dimensional state and a 2-dimensional action space. The actor and critic together contain only 0.009539M trainable parameters, or 0.019078M including the target networks. The RL agent is updated once per unlabeled mini-batch during training and is removed entirely at inference time. Thus, compared with the segmentation framework and frozen foundation models, the RL overhead is negligible in model scale and only affects training.
>
> **Table IV. Average offline prior-generation time per image--text pair.**
> |Metric|Qwen2.5-VL-7B|SAM1|SAM2|
> |---|---:|---:|---:|
> |Avg. time (s/pair)|1.7430|0.4541|0.1661|
>
> [1] FlexMatch: Boosting Semi-Supervised Learning with Curriculum Pseudo Labeling, NIPS 2021.
>
> [2] Adaptive self-training framework for fine-grained scene graph generation, ICLR 2024.

---

> > ### Author Rebuttal · Reviewer_Vp1K · 2026-04-01
> >
> > Thanks for author's rebuttal. After I carefully read, I would raise my score from 3 to 4.

---

> > > ### Author Response · Authors · 2026-04-02
> > >
> > > Dear Reviewer Vp1K,
> > >
> > > We sincerely thank you for your thoughtful comments and for the time and care you invested in re-evaluating our manuscript. We are deeply grateful for your recognition of our additional experiments and rebuttal efforts, as well as for your more positive assessment of our work.
> > >
> > > Your constructive and insightful suggestions mean a great deal to us. They have helped us better understand how to improve the manuscript and have played an important role in strengthening its quality. We will carefully consider each of your comments and address them thoroughly in the revised version.
> > >
> > > Thank you once again for your time and valuable feedback.
> > >
> > >
> > > Sincerely,
> > >
> > > Authors

---

### Decision · Program_Chairs · 2026-04-30

**Decision:**

Accept (regular)

**Comment:**

This paper proposes a learning to label framework for semi-supervised referring expression segmentation that uses a multimodal large language model (MLLM) to initialize soft segmentation proposals and  a reinforcement learning agent to refine and generate pseudo labels. Reviewers consistently acknowledged the clear motivation, innovative aspect of using reinforcement learning for pseudo-label selection, its effectiveness over static thresholding. While the reviewers initially expressed their concerns regarding unclear source of performance gain, complexity of search space, computational complexity in training, robustness to initial noisy pseudo-labels, hallucinations of the MLLM, missing ablations, and not supported claims. In the rebuttal, the authors successfully addressed the major concerns of the reviewers who either raised their scores or maintained their positive stance. The Area Chair supports acceptance and encourages the authors to address these points in the final version.